# Learning Accurate and Interpretable Decision Trees

**Maria-Florina Balcan**[1]                                   **Dravyansh Sharma**[1]

[1]Carnegie Mellon University, Pittsburgh, Pennsylvania, USA

## Abstract

Decision trees are a popular tool in machine learning and yield easy-to-understand models. Several techniques have been proposed in the literature for learning a decision tree classifier, with different techniques working well for data from different domains. In this work, we develop approaches to design decision tree learning algorithms given repeated access to data from the same domain. We propose novel parameterized classes of node splitting criteria in top-down algorithms, which interpolate between popularly used entropy and Gini impurity based criteria, and provide theoretical bounds on the number of samples needed to learn the splitting function appropriate for the data at hand. We also study the sample complexity of tuning prior parameters in Bayesian decision tree learning, and extend our results to decision tree regression. We further consider the problem of tuning hyperparameters in pruning the decision tree for classical pruning algorithms including min-cost complexity pruning. We also study the interpretability of the learned decision trees and introduce a data-driven approach for optimizing the explainability versus accuracy trade-off using decision trees. Finally, we demonstrate the significance of our approach on real world datasets by learning data-specific decision trees which are simultaneously more accurate and interpretable.

## 1   INTRODUCTION

Decision trees are ubiquitous, with applications in operations research, management science, data mining, and machine learning. They are easy to use and understand models that explicitly include the decision rules used in making predictions. Each decision rule is a simple comparsion of a real-valued attribute to a threshold or a categorical attribute against a candidate set of values. Given their remarkable simplicity, decision trees are widely preferred in applications where it is important to justify algorithmic decisions with intuitive explanations Rudin [2018]. However, decades of research on decision trees has resulted in a large suite of candidate approaches for building decision trees Breiman et al. [1984], Mingers [1987], Quinlan [1993, 1996], Kearns and Mansour [1996], Mansour [1997], Maimon and Rokach [2014]. This raises an important question: how should one select the best approach to build a decision tree for the relevant problem domain?

Several empirical studies have been performed comparing various ways to build decision trees Mingers [1989a,b], Esposito et al. [1997], Murthy [1998]. Current wisdom from the literature dictates that for any problem at hand, one needs a domain expert to try out, compare and tune various methods to build the best decision trees for any given problem domain. For instance, the popular Python library Scikit-learn Pedregosa et al. [2011] implements both Gini impurity and entropy as candidate 'splitting criteria' (a crucial component in building the decision trees top-down by deciding which node to split into child nodes), and yet theory suggests another promising candidate Kearns and Mansour [1996] that achieves smaller error bounds under the Weak Hypothesis Assumption[1]. It is therefore desirable to determine which approach works better for the data coming from a given domain. With sufficient data, can we automate this tedious manual process?

In this work we approach this crucial question, and propose ways to build more effective decision trees automatically. Our results show provable learning theoretic guarantees and select methods over larger search spaces than what human experts would typically explore. For example, instead of comparing a small finite number of splitting

---

[1]an *a priori* assumption on the target function. Roughly speaking, it means that the decision tree node functions are already slightly correlated with the target function.

criteria, we examine learnability over continuously infinite parameterized families that yield more effective decision tree learning algorithms.

We consider the problem where the learner has access to multiple related datasets $D_1, \ldots, D_N$ coming from the same problem domain (given by a fixed but unknown distribution $\mathcal{D}$), and the goal is to design a decision tree learning algorithm that works well over the distribution $\mathcal{D}$ using as few datasets ($N$, the sample complexity) as possible. This algorithm design problem is typically formulated as the selection of a hyperparameter from an infinite family. Typically finding the best hyperparameters even on a single problem sample is tedious and computationally intensive, so we would like to bound the number of samples over which we should optimize them, while learning parameters that generalize well over the distribution generating the problem samples. We take steps towards systematically unifying, automating and formalizing the process of designing decision tree learning algorithms, in a way that is adaptive to the data domain.

## 1.1 OUR CONTRIBUTIONS

We formulate the problem of designing a decision tree learning algorithm as a hyperparameter selection problem over multiple problem instances coming from the same domain. Under this formulation, we study the sample complexity, i.e. the number of problem instances needed to learn a provably good algorithm (hyperparameter) under the statistical learning setting (meaning problem instances are drawn from a fixed but unknown distribution) from several different design perspectives important in the construction of decision trees. A key technical challenge is the non-linearity of boundaries of the piecewise structured dual loss function.

- We introduce a novel family of node splitting criterion called $(\alpha, \beta)$-Tsallis entropy criterion, which contains two tunable parameters, and includes several popular node splitting criteria from the literature including the entropy-based ID3/C4.5 Quinlan [1986, 1993] and Gini impurity based CART Breiman et al. [1984]. We bound the sample complexity of provably tuning these hyperparameters in top-down learning algorithms.
- We further study tuning of parameters in Bayesian decision tree learning algorithms used in generating the prior distribution. We also study a parameterized family for node splitting for regression trees and bound the sample complexity of tuning the parameter.
- We next consider the problem of learning the pruning algorithm used in constructing the decision tree. We show how to tune parameters in popular algorithms including the complexity parameter $\tilde{\alpha}$ in the Minimal Cost-Complexity Pruning algorithm, and again obtain sample complexity bounds. We also study the sample complexity of tuning

pessimistic error pruning methods, which are computationally faster.
- We consider the problem of optimizing the explainability-accuracy trade-off in the design of decision tree learning algorithms. Here we consider tuning splitting and pruning parameters simultaneously when growing a decision tree to size $t$ and pruning it down to size $t' \leq t$, while minimizing an objective that incorporates explainability as well as accuracy. Our work is the first to study explainability from a data-driven design perspective.
- We perform experiments to show the practical significance and effectiveness of tuning these hyperparameters on real world datasets.

## 1.2 RELATED WORK

Decision trees Breiman et al. [1984] predate the development of deep learning based methods, but continue to be an extremely popular tool for data analysis and learning simple explainable models. Recent interest in developing interpretable 'white-box' models due to concerns around deployment of deep learning in sensitive and critical decision-making have led to a renewed interest in the study of decision trees Rudin [2019], Loyola-Gonzalez [2019], Molnar [2019]. However, the basic suite of tools for the design of decision trees has seen little advancement over the decades.

*Building and pruning decision trees.* Typically, decision trees are built in two stages. First the tree is grown in a top-down fashion by successively 'splitting' existing nodes according to some *splitting criterion*. Numerous different methods to select which node to split and how to split have been proposed in the literature Breiman et al. [1984], Quinlan [1986, 1993], Kearns and Mansour [1996], Larose and Larose [2014]. The second stage involves pruning the tree to avoid overfitting the training set, and again a variety of approaches are known Breiman et al. [1984], Bohanec and Bratko [1994], Mingers [1987], Quinlan [1987], Mansour [1997]. Furthermore, empirical works suggest that the appropriate method to use, for both splitting and pruning, depends on the data domain at hand Mingers [1989a,b]. The task of selecting the best method or tuning the hyperparameters for a method is left to domain expert. Recent work has developed techniques for computing the optimal decision trees by using branch-and-bound and dynamic programming based techniques Hu et al. [2019], Lin et al. [2020], Demirović et al. [2022]. The key idea is to reduce the search space by tracking bounds on the objective value. However, these approaches are computationally more expensive than the classical greedy methods.

*Tsallis entropy.* Often in modern applications one needs to solve the classification problem over repeated data instances from the same problem domain. In this work, we take steps to automate the process of algorithm selection for decision tree learning using repeated access to data from

the same domain, and also develop more powerful methods for designing decision trees. Our approach is based on Tsallis entropy introduced in the context of statistical physics Tsallis [1988], which has been found to be variously useful in machine learning, for example, as a regularizer in reinforcement learning Chow et al. [2018], Zimmert and Seldin [2021]. Khodak et al. [2023] study tuning of Tsallis entropy in an online meta learning setting for adversarial bandit algorithms. Tsallis entropy based splitting criteria have been empirically studied in the context of decision trees Wang et al. [2016]. We provide a novel two-parameter version that unifies various previously proposed metrics, and provide principled guarantees on the sample complexity of learning the parameters from data.

*Data-driven algorithm design* is a framework for the design of algorithms using machine learning in order to optimize performance over problems coming from a common problem domain Gupta and Roughgarden [2016], Balcan [2020]. The approach has been successful in designing more effective algorithms for a variety of combinatorial problems, ranging from those encountered in machine learning to those in mechanism design Balcan et al. [2018b], Morgenstern and Roughgarden [2015]. The basic premise is to design algorithms for typical inputs instead of worst-case inputs by examining repeated problem instances. In machine learning, this can be used to provably tune hyperparameters Balcan and Sharma [2021], Blum et al. [2021], Bartlett et al. [2022], Balcan et al. [2022a] as opposed to employing heuristics like grid search or random search Bergstra and Bengio [2012] for which formal global-optimality guarantees are typically not known. A key idea is to treat the hyperparameter tuning problem as a statistical learning problem with the parameter space as the hypothesis class and repeated problem samples as data points. Bounding the statistical complexity of this hypothesis class implies sample complexity bounds for hyperparameter tuning using classic learning theory. The framework has formal connections with meta-learning as shown by Balcan et al. [2021b]. Previous work has already shown how to use data-driven algorithm design to improve the the adversarial robustness of non-Lipschitz networks Balcan et al. [2023a] and the running time of branch-and-bound search algorithms Balcan et al. [2018a, 2022b, 2023b]. General techniques have been developed in the latter for providing the sample complexity of tuning a linear combination of variable selection policies in branch-and-bound, and special cases of "path-wise" node selection policies have been studied. In contrast, our work provides new technical insights for node selection policies relevant for decision tree learning which do not satisfy the previously studied path-wise properties and involve a more challenging non-linear interpolation. Prior work Balcan et al. [2021c] obtains a general result for tree search without any path-wise assumptions, but still require a linear interpolation of selection policies.

## 2 PRELIMINARIES AND DEFINITIONS

Let $[k]$ denote the set of integers $\{1, 2, \ldots, k\}$. A (supervised) classification problem is given by a labeled dataset $D = (X, y)$ over some input domain $X \in \mathcal{X}^n$ and $y \in \mathcal{Y}^n = [c]^n$ where $c$ denotes the number of distinct classes or categories. Let $\mathcal{D}$ be a distribution over classification problems of size $n$.[2] We will consider parameterized families of decision tree learning algorithms, parameterized by some parameter $\rho \in \mathcal{P} \subseteq \mathbb{R}^d$ and access to datasets $D_1, \ldots, D_N \sim \mathcal{D}^N$. We do not assume that individual data points $(X_i, y_i)$ are i.i.d. in any dataset $D_j$.

We consider a finite *node function class* $\mathcal{F}$ consisting of boolean functions $\mathcal{X} \to \{0, 1\}$ which are used to label internal nodes in the decision tree, i.e. govern given any data point $x \in \mathcal{X}$ whether the left or right branch should be taken when classifying $x$ using the decision tree. Any given data point $x \in \mathcal{X}$ corresponds to a unique leaf node determined by the node function evaluations at $x$ along some unique root-to-leaf path. Each leaf node of the decision tree is labeled by a class in $[c]$. Given a dataset $(X, y)$ this leaf label is typically set as the most common label for data points $x \in X$ which are mapped to the leaf node.

We denote by $T_{l \to f}$ the tree obtained by *splitting* the leaf node $l$, which corresponds to replacing it by an internal node labeled by $f$ and creating two child leaf nodes. We consider a parameterized class of splitting criterion $\mathcal{G}_\mathcal{P}$ over some parameter space $\mathcal{P}$ consisting of functions $g_\rho : [0, 1]^c \to \mathbb{R}_{\geq 0}$ for $\rho \in \mathcal{P}$. The splitting criterion governs which leaf to be split next and which node function $f \in \mathcal{F}$ to be used when building the decision tree using a top-down learning algorithm which builds a decision tree by successively splitting nodes using $g_\rho$ until the size equals input tree size $t$. More precisely, suppose $w(l)$ (the *weight* of leaf $l$) denotes the number of data points in $X$ that map to leaf $l$, and suppose $p_i(l)$ denotes the fraction of data points labeled by $y = i \in [c]$ among those points that map to leaf $l$. The splitting function over tree $T$ is given by

$$G_\rho(T) = \sum_{l \in \text{leaves}(T)} w(l) g_\rho \left( \{p_i(l)\}_{i=1}^c \right),$$

and we build the decision tree by successively splitting the leaf nodes using node function $f$ which cause the maximum decrease in the splitting function. For example, the information gain criterion may be expressed using $g_\rho(\{p_i(l)\}_{i=1}^c) = -\sum_{i=1}^c p_i \log p_i$.

Algorithm 1 summarizes this well-known general paradigm. We denote the tree obtained by the top-down decision tree learner on dataset $D$ as $T_{\mathcal{F}, \rho, t}(D)$. We study the 0-1 loss of the resulting decision tree classifier. If $T(x) \in [c]$ denotes

---

[2]For simplicity of technical presentation we assume that the dataset size $n$ is fixed across problem instances, but our sample complexity results hold even without this assumption.

**Algorithm 1** Top-down decision tree learner ($\mathcal{F}, g_\rho, t$)

**Input**: Dataset $D = (X, y)$
**Parameters**: Node function class $\mathcal{F}$, splitting criterion $g_\rho \in \mathcal{G}_{\mathcal{P}}$, tree size $t$
**Output**: Decision tree $T$
1: Initialize $T$ to a leaf node labeled by most frequent label $y$ in $D$.
2: **while** $T$ has at most $t$ internal nodes **do**
3:     $l^*, f^* \leftarrow \operatorname{argmin}_{l \in \text{leaves}(T), f \in \mathcal{F}} G_\rho(T_{l \rightarrow f})$
4:     $T \leftarrow T_{l^* \rightarrow f^*}$
5: **return** $T$

the prediction of tree $T$ on $x \in \mathcal{X}$, we define the loss on dataset $D(X, y)$ as

$$L(T, D) := \frac{1}{n} \sum_{i=1}^n \mathbf{I}[T(X_i) \neq y_i],$$

where $\mathbf{I}[\cdot]$ denotes the 0-1 valued indicator function.

# 3 LEARNING TO SPLIT NODES

In this section, we study the sample complexity of learning the splitting criteria. Given a discrete probability distribution $P = \{p_i\}$ with $\sum_{i=1}^c p_i = 1$, we define $(\alpha, \beta)$-Tsallis entropy as

$$g_{\alpha,\beta}^{\text{TSALLIS}}(P) := \frac{C}{\alpha - 1} \left( 1 - \left( \sum_{i=1}^c p_i^\alpha \right)^\beta \right),$$

where $C$ is a normalizing constant (does not affect Algorithm 1), $\alpha \in \mathbb{R}^+, \beta \in \mathbb{Z}^+$. $\beta = 1$ corresponds to standard Tsallis entropy Tsallis [1988]. For example, $\alpha = 2, \beta = 1$ corresponds to Gini impurity, $\alpha = \frac{1}{2}, \beta = 2$ corresponds to the Kearns and Mansour criterion (using which error $\epsilon$ can be achieved with trees of size $\text{poly}(1/\epsilon)$, Kearns and Mansour [1996]) and $\lim_{\alpha \to 1} g_{\alpha,1}^{\text{TSALLIS}}(P)$ yields the (Shannon) entropy criterion. We omit the definitions of these well-known criteria (see Appendix B, proof of Proposition 3.1). Formally, we show in the following proposition that $(\alpha, \beta)$-Tsallis entropy recovers three popular splitting criteria for appropriate values of $\alpha, \beta$.

**Proposition 3.1.** *The splitting criteria $g_{2,1}^{\text{TSALLIS}}(P), g_{\frac{1}{2},2}^{\text{TSALLIS}}(P)$ and $\lim_{\alpha \to 1} g_{\alpha,1}^{\text{TSALLIS}}(P)$ correspond to Gini impurity, the Kearns and Mansour [1996] objective and the entropy criterion respectively.*

We further show that the $g_{\alpha,\beta}^{\text{TSALLIS}}(P)$ family of splitting criteria enjoys the property of being *permissible* splitting criteria (in the sense of Kearns and Mansour [1996]) for any $\alpha \in \mathbb{R}^+, \beta \in \mathbb{Z}^+, \alpha \notin (1/\beta, 1)$, which implies useful desirable guarantees (e.g. ensuring convergences of top-down

learning) for the top-down decision tree learner Kearns and Mansour [1996], De Rosa and Cesa-Bianchi [2015].

**Proposition 3.2.** $(\alpha, \beta)$-*Tsallis entropy has the following properties for any $\alpha \in \mathbb{R}^+, \beta \in \mathbb{Z}^+, \alpha \notin (1/\beta, 1)$*

1. *(Symmetry) For any $P = \{p_i\}$, $Q = \{p_{\pi(i)}$ for some permutation $\pi$ over $[c]\}$, $g_{\alpha,\beta}^{\text{TSALLIS}}(Q) = g_{\alpha,\beta}^{\text{TSALLIS}}(P)$.*

2. $g_{\alpha,\beta}^{\text{TSALLIS}}(P) = 0$ *at any vertex $p_i = 1, p_j = 0$ for all $j \neq i$ of the probability simplex $P$.*

3. *(Concavity) $g_{\alpha,\beta}^{\text{TSALLIS}}(aP + (1-a)Q) \geq a g_{\alpha,\beta}^{\text{TSALLIS}}(P) + (1-a) g_{\alpha,\beta}^{\text{TSALLIS}}(Q)$ for any $a \in [0, 1]$.*

The above properties ensure that $(\alpha, \beta)$-Tsallis entropy is a permissible splitting criterion whenever $\alpha \notin (1/\beta, 1)$. This property makes the $(\alpha, \beta)$-Tsallis entropy an interesting parametric family to study and select the best splitting criterion form, but is not needed for establishing our sample complexity results.

We consider $\alpha \in \mathbb{R}^+$ and $\beta \in [B]$ for some positive integer $B$, and observe that several previously studied splitting criteria can be readily obtained by setting appropriate values of parameters $\alpha, \beta$. We consider the problem of tuning the parameters $\alpha, \beta$ simultaneously when designing the splitting criterion, given access to multiple problem instances (datasets) drawn from some distribution $\mathcal{D}$. The goal is to find parameters $\hat{\alpha}, \hat{\beta}$ based on the training samples, so that on a random $D \sim \mathcal{D}$, the expected loss

$$\mathbb{E}_{D \sim \mathcal{D}} L(T_{\mathcal{F}, (\hat{\alpha}, \hat{\beta}), t}, D)$$

is minimized. We will bound the sample complexity of the ERM Empirical Risk Minimization (ERM) principle, which given $N$ problem samples $D_1, \ldots, D_N$ computes parameters $\hat{\alpha}, \hat{\beta}$ such that

$$\hat{\alpha}, \hat{\beta} = \operatorname{argmin}_{\alpha > 0, \beta \in [B]} \sum_{i=1}^N L(T_{\mathcal{F}, (\alpha, \beta), t}, D_i).$$

We obtain the following guarantee on the sample complexity of learning a near-optimal splitting criterion. The overall argument involves an induction on the size $t$ of the tree (which has appeared in several prior works Megiddo [1978], Balcan et al. [2018a, 2021c, 2022b]), coupled with a counting argument for upper bounding the number of parameter sub-intervals corresponding to different behaviors of Algorithm 1 given a parameter interval corresponding to a fixed partial tree corresponding to an intermediate stage of the algorithm.

**Theorem 3.3.** *Suppose $\alpha > 0$ and $\beta \in [B]$. For any $\epsilon, \delta > 0$ and any distribution $\mathcal{D}$ over problem instances with $n$ examples, $O(\frac{1}{\epsilon^2}(t(\log |\mathcal{F}| + \log t + c \log(B + c)) + \log \frac{1}{\delta}))$ samples drawn from $\mathcal{D}$ are sufficient to ensure that with probability at least $1 - \delta$ over the draw of the samples, the parameters*

$\hat{\alpha}, \hat{\beta}$ learned by ERM over the sample have expected loss that is at most $\epsilon$ larger than the expected loss of the best parameters $\alpha^*, \beta^* = \operatorname{argmin}_{\alpha>0,\beta\geq1}\mathbb{E}_{D\sim\mathcal{D}}L(T_{\mathcal{F},(\hat{\alpha},\hat{\beta}),t}, D)$ over $\mathcal{D}$. Here $t$ is the size of the decision tree, $\mathcal{F}$ is the node function class used to label the nodes of the decision tree and $c$ is the number of label classes.

*Proof Sketch.* Our overall approach is to analyze the structure of the dual class loss function, that is the loss as a function of the hyperparameters $\alpha, \beta$ for a fixed problem instance $(X, y)$. Based on this structure, we give a bound on the pseudodimension of the loss function class which implies a bound on the sample complexity using classic learning theoretic results. In more detail, we show that the loss function is piecewise constant, with a bounded number of pieces by analysing the behavior of Algorithm 1 as the parameters $\alpha, \beta$ are varied. Given this structure, the sample complexity results follow from previously shown bounds on the pseudo-dimension (e.g. Lemma 3.8 of Balcan et al. [2021a]).

Since the loss is completely determined by the final decision tree $T_{\mathcal{F},(\alpha,\beta),t}$, we seek to bound the number of different algorithm behaviors as one varies the hyperparameters $\alpha, \beta$ in Algorithm 1. If the number of internal nodes is $\tau < t$ during the top-down construction, there are $(\tau + 1)|\mathcal{F}|$ choices for $(l, f)$ in Line 3 of Algorithm 1. For any of $\binom{(\tau+1)|\mathcal{F}|}{2}$ pair of candidates $(l_1, f_1)$ and $(l_2, f_2)$, the preference is governed by the splitting functions $G_{\alpha,\beta}(T_{l_1\to f_1})$ and $G_{\alpha,\beta}(T_{l_2\to f_2})$. This preference flips across boundary given by $\sum_{l\in\text{leaves}(T_{l_1\to f_1})} w(l)g_{\alpha,\beta}(\{p_i(l)\}) = \sum_{l\in\text{leaves}(T_{l_2\to f_2})} w(l)g_{\alpha,\beta}(\{p_i(l)\})$. We use the multinomial theorem and Rolle's Theorem to give a bound $O((\beta + c)^c)$ on the number of distinct solutions of the boundary condition for a fixed $\beta$. Over $t$ rounds, this corresponds to at most $O(\Pi_{\tau=1}^t|\mathcal{F}|^2\tau^2(\beta + c)^c)$ critical points across which the algorithmic behaviour (sequence of choices of node splits in Algorithm 1) can change as $\alpha$ is varied for a fixed $\beta$. Adding up over $\beta \in [B]$, we get at most $O(B|\mathcal{F}|^{2t}t^{2t}(B + c)^{ct})$ critical points, which implies a bound of $O(t(\log|\mathcal{F}| + \log t + c\log(B + c))$ on the pseudodimension of the loss function class. This in turn implies the claimed sample complexity guarantee using standard learning theoretic results Anthony and Bartlett [1999], Balcan [2020]. $\square$

Observe that parameter $\alpha$ is tuned over a continuous domain and our near-optimality guarantees hold over the entire continuous domain (as opposed to say over a finite grid of $\alpha$ values). Our results have implications for cross-validation since typical cross-validation can be modeled via a distribution $\mathcal{D}$ created by sampling splits from the same fixed dataset, in which case our results imply how many splits are sufficient to converge to within $\epsilon$ error of best the parameter learned by the cross validation procedure. Similar convergence guarantees have been shown for tuning the

regularization coefficients of the elastic net algorithm for linear regression via cross-validation Balcan et al. [2022a, 2023c]. Our setting is of course more general than just cross validation and includes the case where the different datasets come from related similar tasks for which we seek to learn a common good choice of hyperparameters.

While $(\alpha, \beta)$-Tsallis entropy is well-motivated as a parameterized class of node splitting criteria as it includes several previously studied splitting criteria, and generalizes the Tsallis entropy which may be of independent interest in other applications, it involves simulatenous optimization of two parameters which can be computationally challenging. To this end, we define the following single parameter family which interpolates known node splitting methods:

$$g_\gamma(\{p_i\}) := C\left(\Pi_i p_i\right)^\gamma,$$

where $\gamma \in (0, 1]$ and $C$ is some constant. For binary classification, the setting $\gamma = \frac{1}{2}$ and $\gamma = 1$ correspond to Kearns and Mansour [1996] and Gini impurity respectively, for appropriate choice of $C$. It is straightforward to verify that $g_\gamma$ is permissible for all $\gamma \in (0, 1]$, i.e. is symmetric, zero at simplical vertices and concave. We show the following improved sample complexity guarantee for tuning $\gamma$ (proof in Appendix B). Note that this family is not a special case of $(\alpha, \beta)$-Tsallis entropy, but contains additional splitting functions which may work well on given domain-specific data. Also, since it has a single parameter, it can be easier to optimize efficiently in practice.

**Theorem 3.4.** *Suppose $\gamma \in (0, 1]$. For any $\epsilon, \delta > 0$ and any distribution $\mathcal{D}$ over problem instances with $n$ examples, $O(\frac{1}{\epsilon^2}(t(\log|\mathcal{F}| + \log t) + \log\frac{1}{\delta}))$ samples drawn from $\mathcal{D}$ are sufficient to ensure that with probability at least $1 - \delta$ over the draw of the samples, the parameter $\hat{\gamma}$ learned by ERM over the sample is $\epsilon$-optimal, i.e. has expected loss at most $\epsilon$ larger than that of the optimal parameter over $\mathcal{D}$.*

## 3.1 BAYESIAN DECISION TREE MODELS

Several Bayesian approaches for building a decision tree have been proposed in the literature Chipman et al. [1998, 2002], Wu et al. [2007]. The key idea is to specify a prior which induces a posterior distribution and a stochastic search is performed using Metropolis-Hastings algorithms to explore the posterior and find an effective tree. We will summarize the overall approach below and consider the problem of tuning parameters in the prior, which control the accuracy and size of the tree. Unlike most of prior research on data-driven algorithm design which study deterministic algorithms, we will analyze the learnability of parameters in a randomized algorithm. One notable exception is the study of random initialization of centers in $k$-center clustering via parameterized Llyod's families Balcan et al. [2018b].

**$\sigma, \phi$-Bayesian algorithm family.** Let $F = (f_1, \ldots, f_t)$ denote the node functions at the nodes of the decision tree $T$. The prior $p(F, T)$ is specified using the relationship

$$p(F, T) = p(F|T)p(T).$$

We start with a tree $T$ consisting of a single root node. For any node $\tau$ in $T$, it is split with probability $p_{\text{SPLIT}}(\tau) = \sigma(1 + d_\tau)^{-\phi}$, and if split, the process is repeated for the left and right children. Here $d_\tau$ denotes the depth of node $\tau$, and $\sigma, \phi$ are hyperparameters. The size of generated tree is capped to some upper bound $t$. Intuitively, $\sigma$ controls the size of the tree and $\phi$ controls its depth. At each node, the node function is selected uniformly at random from $\mathcal{F}$. This specifies the prior $p(T)$. The conjugate prior for the node functions $F = (f_1, \ldots, f_t)$ is given by the standard Dirichlet distribution of dimension $c - 1$ (recall $c$ is the number of label classes) with parameter $a = (a_1, \ldots, a_c), a_i > 0$. Under this prior, the label predictions are given by

$$p(y \mid X, T) = \left( \frac{\Gamma(\sum_i a_i)}{\Pi_i \Gamma(a_i)} \right)^t \prod_{j=1}^t \frac{\Pi_i \Gamma(n_{ji} + a_i)}{\Gamma(n_j + \sum_i a_i)},$$

where $n_{ji} = \sum_k \mathbf{I}(y_{jk} = i)$ counts the number of datapoints with label $i$ at node $j$, $n_j = \sum_i n_{ji}$ and $i = 1, \ldots, c$. $a$ is usually set as the vector $(1, \ldots, 1)$ which corresponds to the uniform Dirichlet prior. Finally the stochastic search of the induced posterior is done using the Metropolis-Hastings (MH) algorithm for simulating a Markov chain Chipman et al. [1998]. Starting from a single root node, the initial tree $T^0$ is grown according to the prior $p(T)$. Then to construct $T^{i+1}$ from $T^i$, a new tree $T^*$ is constructed by splitting a random node using a random node function, pruning a random node, reassigning a node function or swapping the node functions of a parent and a child node. Then we set $T^{i+1} = T^*$ with probability $q(T^i, T^*)$ according to the posterior $p(y \mid X, T)$, or keep $T^{i+1} = T^i$ otherwise. The algorithm outputs the tree $T^\omega$ where $\omega$ is typically a fixed large number of iterations (say 10000) to ensure that the search space is explored sufficiently well.

**Hyperparameter tuning.** We consider the problem of tuning of prior hyperparameters $\sigma, \phi$, to obtain the best expected performance of the algorithm. To this end, we define $\mathbf{z} = (\mathbf{z}_1, \ldots, \mathbf{z}_{t-1}) \in [0, 1]^{t-1}$ as the randomness used in generating the tree $T$ according to $p(T)$. Let $T_{\mathbf{z}, \sigma, \phi}$ denote the resulting initial tree. Let $\mathbf{z}'$ denote the remaining randomness used in the selecting the random node function and the stochastic search, resulting in the final tree $T(T_{\mathbf{z}, \sigma, \phi}, \mathbf{z}', \omega)$. Our goal is to learn the hyperparameters $\sigma, \phi$ which minimize the expected loss

$$\mathbb{E}_{\mathbf{z}, \mathbf{z}', \mathcal{D}} L(T(T_{\mathbf{z}, \sigma, \phi}, \mathbf{z}', \omega), D),$$

where $\mathcal{D}$ denotes the distribution according to which the data $D$ is sampled, and $L$ denotes the expected fraction of incorrect predictions by the learned Bayesian decision tree. ERM over a sample $D_1, \ldots, D_n \sim \mathcal{D}^n$ finds the parameters $\hat{\sigma}, \hat{\phi}$ which minimize the expected average loss $\frac{1}{n} \sum_{i=1}^n \mathbb{E}_{\mathbf{z}, \mathbf{z}'} L(T(T_{\mathbf{z}, \sigma, \phi}, \mathbf{z}', \omega), D_i)$ over the problem instances in the sample. It is not clear how to efficiently implement this procedure. However, we can bound its sample complexity and prove the following guarantee for learning a near-optimal prior for the Bayesian decision tree.

**Theorem 3.5.** *Suppose $\sigma, \phi > 0$. Consider the problem of designing a Bayesian decision tree learning algorithm by selecting the parameters from the $\sigma, \phi$-Bayesian algorithm family. For any $\epsilon, \delta > 0$ and any distribution $\mathcal{D}$ over problem instances with $n$ examples, $O(\frac{1}{\epsilon^2}(\log t + \log \frac{1}{\delta}))$ samples drawn from $\mathcal{D}$ are sufficient to ensure that with probability at least $1 - \delta$ over the draw of the samples, the parameters $\hat{\sigma}, \hat{\phi}$ learned by ERM over the sample have expected loss that is at most $\epsilon$ larger than the expected loss of the best parameters. Here $t$ denotes an upper bound on the size of the decision tree.*

*Proof Sketch.* Fix the dataset $D$ and fix the randomness $\mathbf{z}$ used to generate the tree $T$. We example the two-dimensional functional curve in the $\sigma$-$\phi$ parameter space, across which the generated tree $T$ changes due to a change in the splitting decision. We show that across $N$ problem instances, there are a total of at most $O(t^2 N^2)$ pieces of the loss function where distinct trees are generated across the instances. We use this piecewise loss structure to bound the Rademacher complexity, which in turn implies uniform convergence guarantees by applying standard learning-theoretic results. $\square$

So far, we have considered learning decision tree classifiers that classify any given data point into one of finitely many label classes. In the next subsection, we consider an extension of the setting to learning over regression data, for which decision trees are again known as useful interpretable models Breiman et al. [1984].

## 3.2 SPLITTING REGRESSION TREES

In the regression problem, we have $\mathcal{Y} = \mathbb{R}$ and the top-down learning algorithm can still be used but with continous splitting criteria. Popular splitting criteria for regression trees include the mean squared error (MSE) and half Poisson deviance (HPD). Let $y_l$ denote the set of labels for data points classified by leaf node $l$ in tree $T$ $\overline{y_l} := \frac{1}{|y_l|} \sum_{y \in y_l} y$ is the mean prediction for node $l$. MSE is defined as $g_{\text{MSE}}(y_l) := \frac{1}{|y_l|} \sum_{y \in y_l} (y - \overline{y_l})^2$ and HPD as $g_{\text{HPD}}(y_l) := \frac{1}{|y_l|} \sum_{y \in y_l} (y \log \frac{y}{\overline{y_l}} - y + \overline{y_l})$. These are interpolated by the mean Tweedie deviance Zhou et al. [2022] error with power $p$ given by

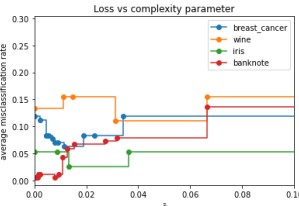

Figure 1: The loss of pruned tree as a function of the minimum cost-complexity pruning parameter $\tilde{\alpha}$ is piecewise constant with at most $t$ pieces. The optimal complexity parameter $\tilde{\alpha}$ varies with dataset.

$$g_p(y_l) := \frac{2}{|y_l|} \sum_{y \in y_l} \left( \frac{\max\{y, 0\}^{2-p}}{(1-p)(2-p)} - \frac{y\overline{y_l}}{1-p} + \frac{\overline{y_l}^{2-p}}{2-p} \right),$$

where $p = 0$ corresponds to MSE and the limit $p \to 1$ corresponds to HPD. We call this the $p$-Tweedie splitting criterion, and have the following sample complexity guarantee for tuning $p$ in the multiple instance setting.

**Theorem 3.6.** *Suppose $p \in [0, 1]$. For any $\epsilon, \delta > 0$ and any distribution $\mathcal{D}$ over problem instances with $n$ examples, $O(\frac{1}{\epsilon^2}(t(\log|\mathcal{F}| + n) + \log\frac{1}{\delta}))$ samples drawn from $\mathcal{D}$ are sufficient to ensure that with probability at least $1 - \delta$ over the draw of the samples, the Tweedie power parameter $\hat{p}$ learned by ERM over the sample is $\epsilon$-optimal. $\mathcal{F}$ here the node function class, assumed to be finite (Section 2).*

Since $t < n$, this indicates that tuning regression parameters typically (for sufficiently small $B, c$) has a larger sample complexity upper bound.

## 4 LEARNING TO PRUNE

Some leaf nodes in a decision tree learned via the top-down learning algorithm may involve nodes that overfit to a small number of data points. This overfitting problem in decision tree learning is typically resolved by pruning some of the branches and reducing the tree size Breiman et al. [1984]. The process of growing trees to size $t$ and pruning back to smaller size $t'$ tends to produce more effective decision trees than learning a tree of size $t'$ top-down. We study the minimum cost-complexity pruning algorithm here, which involves a tunable complexity parameter $\tilde{\alpha}$, and establish bounds on the sample complexity of tuning $\tilde{\alpha}$ given access to repeated problem instances from dataset distribution $\mathcal{D}$.

The cost-complexity function for a tree $T$ is given by

$$R(T, D) := L(T, D) + \tilde{\alpha}|\text{leaves}(T)|.$$

More leaf nodes correspond to higher flexibility of the decision tree in partitioning the space into smaller pieces and

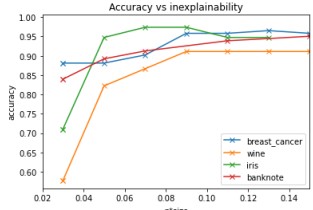

Figure 2: Accuracy vs $\eta * |\text{leaves}(T)|$ as the pruning parameter $\tilde{\alpha}$ is varied, for $\eta = 0.01$.

therefore greater ability to fit the training data. $\tilde{\alpha} \in [0, \infty)$ controls how strongly we penalize this increased complexity of the tree. The minimum cost-complexity pruning algorithm computes a subtree $T_{\tilde{\alpha}}$ of $T$ which minimizes the cost-complexity function. When $\tilde{\alpha} = 0$, this selects $T$ and when $\tilde{\alpha} = \infty$ a single node tree is selected.

Given a leaf node $l$ of $T$ labeled by $i \in [c]$, the cost-complexity measure is defined to be $R(l, D) = \frac{w(l) - p_i(l)}{w(l)} + \tilde{\alpha}$. Denote by $T_t$, the branch of tree $T$ rooted at node $t$ and $R(T_t, D) := \sum_{l \in \text{leaves}(T_t)} R(l, D) + \tilde{\alpha}|\text{leaves}(T_t)|$. The minimum cost-complexity pruning algorithm successively deletes weakest links which minimize $\frac{R(t, D) - R(T_t, D)}{|\text{leaves}(T_t)| - 1}$ over internal nodes $t$ of the currently pruned tree.

We have the following result bounding the sample complexity of tuning $\tilde{\alpha}$ from multiple data samples. Proof details of results in this section are located in Appendix C.

**Theorem 4.1.** *Suppose $\tilde{\alpha} \in \mathbb{R}_{\geq 0}$ and $t$ denote the size of the unpruned tree. For any $\epsilon, \delta > 0$ and any distribution $\mathcal{D}$ over problem instances with $n$ examples, $O(\frac{1}{\epsilon^2}(\log t + \log\frac{1}{\delta}))$ samples drawn from $\mathcal{D}$ are sufficient to ensure that with probability at least $1 - \delta$ over the draw of the samples, the minimum cost-complexity pruning parameter learned by ERM over the sample is $\epsilon$-optimal.*

Minimum cost-complexity pruning Breiman et al. [1984] can be implemented using a simple dynamic program to find the sequence of trees that minimize $R(T, D)$ for any given fixed $\tilde{\alpha}$, which takes quadratic time to implement in the size of $T$ Bohanec and Bratko [1994]. Faster pruning approaches are known that directly prune nodes for which the reduction in error or splitting criterion when splitting the node is not statistically significant. This includes Critical Value Pruning Mingers [1987, 1989a] and Pessimistic Error Pruning Quinlan [1987]. Principled statistical learning guarantees are known for the latter Mansour [1997], and here we will consider the problem of tuning the confidence parameter in pessimistic pruning, which we describe below.

Suppose $\mathcal{X} \subseteq \mathbb{R}^a$, i.e. each data point consists of $a$ real features or attributes. For any internal node $h$ of $T$, if $e_h$ denotes the fraction of data points that are misclassified among the $n_h$ data points that are classified via the sub-tree rooted at $h$, and $e_l$ denotes the fraction of misclassified data

points if $h$ is replaced by a leaf node, then the pessimistic pruning test of Mansour [1997] is given by

$$e_l \leq e_h + c_1 \sqrt{\frac{t_h \log a + c_2}{n_h}},$$

where $c_1$ and $c_2$ are parameters, and $t_h$ denotes the size of the sub-tree rooted at $h$. We consider the problem of tuning $c_1, c_2$ given repeated data samples, and bound the sample complexity of tuning in the following theorem.

**Theorem 4.2.** *Suppose $c_1, c_2 \in \mathbb{R}_{\geq 0}$ and $t$ denote the size of the unpruned tree. For any $\epsilon, \delta > 0$ and any distribution $\mathcal{D}$ over problem instances with $n$ examples, $O(\frac{1}{\epsilon^2}(\log t + \log \frac{1}{\delta}))$ samples drawn from $\mathcal{D}$ are sufficient to ensure that with probability at least $1 - \delta$ over the draw of the samples, the pessimistic pruning parameters learned by ERM over the sample is $\epsilon$-optimal.*

We have studied parameter tuning in two distinct parameterized approaches for decision tree pruning. However, several other pruning methods are known in the literature Esposito et al. [1997, 1999], and it is an interesting direction for future research to design approaches to select the best method based on data. We conclude this section with a remark about another interesting future direction, namely extending our results to tree ensembles.

**Remark 1** (Extension to tree ensembles). *Extension of our approaches to tree ensembles is an interesting question, although this comes at the expense of making the model less interpretable. We still need to choose splitting and pruning methods used in building the individual trees. If we learn a uniform splitting criterion for all trees, our sample complexity arguments are straightforward to extend to this case and would imply an additional $O(n_t)$ factor in the sample complexity, where $n_t$ is the number of trees in the random forest (in the case of pruning, our arguments would imply an $O(\log n_t)$ term). There are interesting further questions here, including learning a combination of splitting/pruning criteria across different trees and tuning the number of trees $n_t$ as a hyperparameter (which impacts both accuracy and interpretability).*

## 5 OPTIMIZING THE EXPLAINABILITY VERSUS ACCURACY TRADE-OFF

Decision trees are often regarded as one of the preferred models when the model predictions need to be explainable. Complex or large decision trees can however not only overfit the data but also hamper model interpretability. So far we have considered parameter tuning when building or pruning the decision tree with the goal of optimizing accuracy on unseen "test" datasets on which the decision tree is built using the learned hyperparameters. We will consider a modified objective here which incorporates model complexity in

the test objective. That is, we seek to find hyperparameters $\alpha, \beta, \tilde{\alpha}$ based on the training samples, so that on a random $D \sim \mathcal{D}$, the expected loss

$$L_\eta := \mathbb{E}_{D \sim \mathcal{D}} L(T, D) + \eta |\text{leaves}(T)|$$

is minimized, where $\eta \geq 0$ is the complexity coefficient. This objective has been studied in a recent line of work which designs techniques for provably optimal decision trees with high interpretability Hu et al. [2019], Lin et al. [2020]. Note that, while the objective is similar to min cost-complexity pruning, there the regularization term $\tilde{\alpha} |\text{leaves}(T)|$ is added to the training objective to get the best generalization accuracy on test data. In contrast, we add the regularization term to the test objective itself and $\eta$ here is a fixed parameter that governs the balance between accuracy and explainability that the learner aims to strike.

Our approach here is to combine tunable splitting and pruning to optimize the accuracy-explainability trade-off. We set $(\alpha, \beta)$-Tsallis entropy as the splitting criterion and min cost-complexity pruning with parameter $\tilde{\alpha}$ as the pruning algorithm. We show the following upper bound on the sample complexity when simultaneously learning to split and prune.

**Theorem 5.1.** *Suppose $\alpha > 0, \beta \in [B], \tilde{\alpha} \geq 0$. For any $\epsilon, \delta > 0$ and any distribution $\mathcal{D}$ over problem instances with $n$ examples, $O(\frac{1}{\epsilon^2}(t(\log |\mathcal{F}| + \log t + c \log(B + c)) + \log \frac{1}{\delta}))$ samples drawn from $\mathcal{D}$ are sufficient to ensure that with probability at least $1 - \delta$ over the draw of the samples, the parameters learned by ERM for $L_\eta$ are $\epsilon$-optimal.*

## 6 EXPERIMENTS

We examine the significance of the novel splitting techniques and the importance of designing data-driven decision tree learning algorithms via hyperparameter tuning for various benchmark datasets. We only perform small-scale simulations that can be run on a personal computer and include code in the supplementary material for reproducibility. The datasets used are from the UCI repository, are publicly available and are briefly described below.

*Iris* Fisher [1936] consists of three classes of the iris plant and four real-valued attributes. A total of 150 instances, 50 per class. *Wine* Lichman et al. [2013] has three classes of wines, 13 real attributes and 178 data points in all. *Breast cancer (Wisconsin diagnostic)* contains 569 instances, with 30 features, and two classes, malignant and benign Wolberg et al. [1994]. The *Banknote Authentication* dataset Lohweg [2013] also involves binary classification and has 1372 data points and five real attributes. These datasets are selected to capture a variety of attribute sizes and number of data points.

We first study the effect of choice of $(\alpha, \beta)$ parameters in the Tsallis entropy based splitting criterion. For each dataset,

| Dataset | Best $(\alpha^*, \beta^*)$ | Acc$(\alpha^*, \beta^*)$ | Acc(Gini) | Acc(Entropy) | Acc(KM96) |
|---|---|---|---|---|---|
| Iris | (0.5,1) | $96.00 \pm 1.85$ | $92.99 \pm 1.53$ | $93.33 \pm 1.07$ | $94.67 \pm 2.70$ |
| Banknote | (2.45,2) | $98.32 \pm 0.52$ | $97.01 \pm 0.59$ | $97.30 \pm 1.62$ | $97.00 \pm 1.79$ |
| Breast cancer | (0.5, 3) | $94.69 \pm 0.77$ | $92.92 \pm 1.29$ | $93.01 \pm 1.05$ | $93.27 \pm 1.16$ |
| Wine | (2.15,6) | $96.57 \pm 1.88$ | $89.14 \pm 3.18$ | $92.57 \pm 2.38$ | $93.71 \pm 2.26$ |

Table 1: A comparison of the performance of different splitting criteria. The first column indicates the best $(\alpha, \beta)$ parameters for each dataset over the grid considered in Figure 3. Acc denotes test accuracy along with a 95% confidence interval.

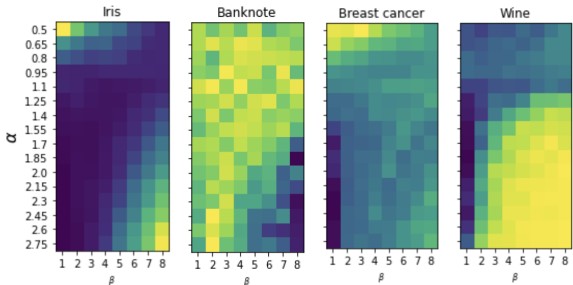

Figure 3: Average test accuracy (proportional to brightness, yellow is highest) of $(\alpha, \beta)$-Tsallis entropy based splitting criterion as the parameters are varied, across datasets. We observe that different parameter settings work best for each dataset, highlighting the need to learn data-specific values.

we perform 5-fold cross validation for a large grid of parameters depicted in Figure 3 and measure the accuracy on held out test set consisting of 20% of the datapoints (i.e. training datasets are just random subsets of the 80% of the dataset used for learning the parameters). We implement a slightly more sophisticated variant of Algorithm 1 which grows the tree to maximum depth of 5 (as opposed to a fixed size $t$). We do not use any pruning here. There is a remarkable difference in the optimal parameter settings for different datasets. Moreover, we note in Table 1, that carefully chosen values of $(\alpha, \beta)$ significantly outperform standard heuristics like Gini impurity or entropy based splitting, or even specialized heuristics like Kearns and Mansour [1996] for which worst-case error guarantees (assuming weak learning) are known. This further underlines the significance of data-driven algorithm design for decision tree learning.

We further study the impact of tuning the complexity paramter $\tilde{\alpha}$ in the minimum cost-complexity pruning algorithm. The test error varies with $\tilde{\alpha}$ in a data dependent way and different data could have different optimal parameter as depicted in Figure 1. We use Gini impurity as the splitting criterion. Furthermore, we observe that on a single instance, the average test error is a piecewise constant function with at most $t$ pieces which motivates the sample complexity bound in Theorem 4.1.

We also examine the explainability-accuracy trade-off as given by our regularized objective with complexity coefficient $\eta$. In Figure 2, we plot the explainability-accuracy frontier as the pruning parameter $\tilde{\alpha}$ is varied. Here we fix the splitting criterion as the Gini impurity. For a given dataset, this frontier can be pushed by a careful choice of the splitting criterion (Theorem 5.1). We defer these examinations and further experiments to the appendix.

# 7 CONCLUSION

We consider the problem of automatically designing decision tree learning algorithms by data-driven selection of hyperparameters. Previous extensive research has observed that different ways to split or prune nodes when building a decision tree work best for data coming from different domain. We present a novel splitting criterion called $(\alpha, \beta)$-Tsallis entropy which interpolates popular previously known methods into a rich infinite class of algorithms. We consider the setting where we have repeated access to data from the same domain and provide formal bounds on the sample complexity of tuning the hyperparameters for the ERM principle. We extend our study to learning regression trees, selecting pruning parameters, and optimizing over the explainability-accuracy trade-off. Empirical simulations validate our theoretical study and highlight the significance and usefulness of learning decision tree algorithms.

Our work presents several directions for future research. While our results provide guarantees on sample efficiency, the problem of computationally efficient optimization of the sample accuracy is left open. Another direction for future research is designing and analyzing a potentially more powerful algorithm family for pruning, and extending our results to tree ensembles. We also remark that we focus on upper bounds on sample complexity, and providing corresponding lower bounds is an interesting avenue for further research.

**Acknowledgements**

We thank Avrim Blum, Misha Khodak, Hedyeh Beyhaghi, Siddharth Prasad and Keegan Harris for helpful comments. This material is based on work supported by the National Science Foundation under grants CCF1910321, IIS 1901403, and SES 1919453; and the Defense Advanced Research Projects Agency under cooperative agreement HR00112020003.

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

# Learning Accurate and Interpretable Decision Trees
## (Supplementary Material)

**Maria-Florina Balcan**[1]                    **Dravyansh Sharma**[1]

[1]Carnegie Mellon University, Pittsburgh, Pennsylvania, USA

## A    STANDARD RESULTS FROM LEARNING THEORY AND DATA-DRIVEN ALGORITHM DESIGN

The pseudo-dimension is frequently used to analyze the learning theoretic complexity of real-valued function classes. The formal definition is stated here for convenience.

**Definition 1** (Shattering and Pseudo-dimension, Anthony and Bartlett [1999]). *Let $\mathcal{F}$ be a set of functions mapping from $\mathcal{X}$ to $\mathbb{R}$, and suppose that $S = \{x_1, \ldots, x_m\} \subseteq \mathcal{X}$. Then $S$ is pseudo-shattered by $\mathcal{F}$ if there are real numbers $r_1, \ldots, r_m$ such that for each $b \in \{0,1\}^m$ there is a function $f_b$ in $\mathcal{F}$ with $\text{sign}(f_b(x_i) - r_i) = b_i$ for $i \in [m]$. We say that $r = (r_1, \ldots, r_m)$ witnesses the shattering. We say that $\mathcal{F}$ has pseudo-dimension $d$ if $d$ is the maximum cardinality of a subset $S$ of $\mathcal{X}$ that is pseudo-shattered by $\mathcal{F}$, denoted $\text{Pdim}(\mathcal{F}) = d$. If no such maximum exists, we say that $\mathcal{F}$ has infinite pseudo-dimension.*

Pseudo-dimension is a real-valued analogue of VC-dimension, and is a classic complexity notion in learning theory due to the following theorem which implies the uniform convergence sample complexity for any function in class $\mathcal{F}$ when $\text{Pdim}(\mathcal{F})$ is finite.

**Theorem A.1** (Uniform convergence sample complexity via pseudo-dimension, Anthony and Bartlett [1999]). *Suppose $\mathcal{H}$ is a class of real-valued functions with range in $[0, H]$ and finite $\text{Pdim}(\mathcal{F})$. For every $\epsilon > 0$ and $\delta \in (0, 1)$, the sample complexity of $(\epsilon, \delta)$-uniformly learning the class $\mathcal{H}$ is $O\left(\left(\frac{H}{\epsilon}\right)^2 \left(\text{Pdim}(\mathcal{F}) \log\left(\frac{H}{\epsilon}\right) + \log\left(\frac{1}{\delta}\right)\right)\right)$.*

Uniform learning is closely related to the notion of PAC (probably approximately correct) learning, indeed $(\epsilon, \delta)$-uniform learning corresponds to $(\epsilon/2, \delta)$-PAC learning Mohri et al. [2018].

We also need the following lemma from data-driven algorithm design.

**Lemma A.2.** *(Lemma 2.3, Balcan [2020], Lemma 3.8 Balcan et al. [2021a]) Suppose that for every problem instance $D \in \mathbf{D}$, the function $L_D(\rho): \mathbb{R} \to \mathbb{R}$ is piecewise constant with at most $N$ pieces. Then the family $\{L_\rho(\cdot)\}$ over instances in $\mathbf{D}$ has pseudo-dimension $O(\log N)$.*

The follwing theorem is due to Bartlett et al. [2022] and is useful in obtaining some of our pseudodimension bounds.

**Theorem A.3** (Bartlett et al. [2022]). *Suppose that each function $f \in \mathcal{F}$ is specified by $n$ real parameters. Suppose that for every $x \in \mathcal{X}$ and $r \in \mathbb{R}$, there is a GJ algorithm $\Gamma_{x,r}$ that given $f \in \mathcal{F}$, returns "true" if $f(x) \geq r$ and "false" otherwise. Assume that $\Gamma_{x,r}$ has degree $\Delta$ and predicate complexity $\Lambda$. Then, $\text{Pdim}(\mathcal{F}) = O(n \log(\Delta\Lambda))$.*

## B    PROOFS FROM SECTION 3

**Proposition 3.1 (restated)** *The splitting criteria $g_{2,1}^{\text{TSALLIS}}(P)$, $g_{\frac{1}{2},2}^{\text{TSALLIS}}(P)$ and $\lim_{\alpha \to 1} g_{\alpha,1}^{\text{TSALLIS}}(P)$ correspond to Gini impurity, the Kearns and Mansour [1996] objective and the entropy criterion respectively.*

*Proof of Proposition 3.1.* Setting $\alpha = 2, \beta = 1$ immediately yields the expression for Gini impurity. Plugging $\alpha = \frac{1}{2}, \beta = 2$ yields

$$
g_{\frac{1}{2},2}^{\text{TSALLIS}}(P) = \frac{C}{-\frac{1}{2}} \left( 1 - \left( \sum_{i=1}^{c} \sqrt{p_i} \right)^2 \right)
$$

$$
= 2C \left( \sum_{i=1}^{c} p_i + 2 \sum_{i \neq j} \sqrt{p_i p_j} - 1 \right)
$$

$$
= 4C \sum_{i \neq j} \sqrt{p_i p_j}.
$$

For $c = 2$, $g_{\frac{1}{2},2}^{\text{TSALLIS}}(P) = 4C \sqrt{p_1(1 - p_1)}$ which matches the splitting function of Kearns and Mansour [1996]. Also taking the limit $\alpha \to 1$ gives

$$
g_{\alpha \to 1, \beta}^{\text{TSALLIS}}(P) = \lim_{\alpha \to 1} \frac{C}{\alpha - 1} \left( 1 - \left( \sum_{i=1}^{c} p_i^{\alpha} \right)^{\beta} \right)
$$

$$
= -C\beta \left( \sum_{i=1}^{c} p_i^{\alpha} \right)^{\beta - 1} \left( \sum_{i=1}^{c} p_i^{\alpha} \ln p_i \right)
$$

$$
= -C\beta \left( \sum_{i=1}^{c} p_i \ln p_i \right).
$$

For $\beta = 1$, this corresponds to the entropy criterion. $\qquad\square$

**Proposition 3.2 (restated)** $(\alpha, \beta)$-*Tsallis entropy has the following properties for any* $\alpha \in \mathbb{R}^+, \beta \in \mathbb{Z}^+, \alpha \notin (1/\beta, 1)$

1. *(Symmetry) For any* $P = \{p_i\}$, $Q = \{p_{\pi(i)}$ *for some permutation* $\pi$ *over* $[c]$, $g_{\alpha,\beta}^{\text{TSALLIS}}(Q) = g_{\alpha,\beta}^{\text{TSALLIS}}(P)$.
2. $g_{\alpha,\beta}^{\text{TSALLIS}}(P) = 0$ *at any vertex* $p_i = 1, p_j = 0$ *for all* $j \neq i$ *of the probability simplex* $P$.
3. *(Concavity)* $g_{\alpha,\beta}^{\text{TSALLIS}}(aP + (1 - a)Q) \geq a g_{\alpha,\beta}^{\text{TSALLIS}}(P) + (1 - a) g_{\alpha,\beta}^{\text{TSALLIS}}(Q)$ *for any* $a \in [0, 1]$.

*Proof of Proposition 3.2.* Properties 1 and 2 are readily verified. We further show that $(\alpha, \beta)$-Tsallis entropy is concave for $\alpha, \beta > 0, \alpha\beta \geq 1$.

First consider the case $\alpha \geq 1$. We use the fact that the univariate function $f(x) = x^{\theta}$ is convex for all $\theta \geq 1$. For any $a \in [0, 1]$, $P = \{p_i\}_{i=1}^{c}, Q = \{q_i\}_{i=1}^{c}$,

$$
g_{\alpha,\beta}^{\text{TSALLIS}}(aP + (1 - a)Q) = \frac{C}{\alpha - 1} \left( 1 - \left( \sum_{i=1}^{c} (ap_i + (1 - a)q_i)^{\alpha} \right)^{\beta} \right)
$$

$$
\geq \frac{C}{\alpha - 1} \left( 1 - \left( \sum_{i=1}^{c} ap_i^{\alpha} + (1 - a)q_i^{\alpha} \right)^{\beta} \right)
$$

$$
= \frac{C}{\alpha - 1} \left( 1 - \left( a \sum_{i=1}^{c} p_i^{\alpha} + (1 - a) \sum_{i=1}^{c} q_i^{\alpha} \right)^{\beta} \right)
$$

$$
\geq \frac{C}{\alpha - 1} \left( 1 - \left( a \left( \sum_{i=1}^{c} p_i^{\alpha} \right)^{\beta} + (1 - a) \left( \sum_{i=1}^{c} q_i^{\alpha} \right)^{\beta} \right) \right)
$$

$$
= a g_{\alpha,\beta}^{\text{TSALLIS}}(P) + (1 - a) g_{\alpha,\beta}^{\text{TSALLIS}}(Q).
$$

It remains to consider the case $0 < \alpha \leq 1/\beta$. In this case, we apply the reverse Minkowski's inequality and use that $\alpha\beta \leq 1$ to establish concavity.

$$g_{\alpha,\beta}^{\text{TSALLIS}}(aP + (1-a)Q) = \frac{C}{\alpha-1}\left(1 - \left(\sum_{i=1}^{c}(ap_i + (1-a)q_i)^{\alpha}\right)^{\beta}\right)$$

$$\geq \frac{C}{\alpha-1}\left(1 - \left(\left(\sum_{i=1}^{c}(ap_i)^{\alpha}\right)^{\frac{1}{\alpha}} + \left(\sum_{i=1}^{c}((1-a)q_i)^{\alpha}\right)^{\frac{1}{\alpha}}\right)^{\alpha\beta}\right)$$

$$= \frac{C}{\alpha-1}\left(1 - \left(a\left(\sum_{i=1}^{c}p_i{}^{\alpha}\right)^{\frac{1}{\alpha}} + (1-a)\left(\sum_{i=1}^{c}q_i^{\alpha}\right)^{\frac{1}{\alpha}}\right)^{\alpha\beta}\right)$$

$$\geq \frac{C}{\alpha-1}\left(1 - \left(a\left(\sum_{i=1}^{c}p_i{}^{\alpha}\right)^{\frac{1}{\alpha}\cdot\alpha\beta} + (1-a)\left(\sum_{i=1}^{c}q_i^{\alpha}\right)^{\frac{1}{\alpha}\cdot\alpha\beta}\right)\right)$$

$$= ag_{\alpha,\beta}^{\text{TSALLIS}}(P) + (1-a)g_{\alpha,\beta}^{\text{TSALLIS}}(Q).$$

$\square$

To prove Theorem 3.3, we state below a simple helpful lemma, which is a simple consequence of the Rolle's Theorem.

**Lemma B.1** (e.g. Lemma 26 in Balcan and Sharma [2021]). *The equation $\sum_{i=1}^{n}a_i e^{b_i x} = 0$ where $a_i, b_i \in \mathbb{R}$ has at most $n-1$ distinct solutions $x \in \mathbb{R}$.*

We will now restate and prove Theorem 3.3.

**Theorem 3.3 (restated)** *Suppose $\alpha > 0$ and $\beta \in [B]$. For any $\epsilon, \delta > 0$ and any distribution $\mathcal{D}$ over problem instances with $n$ examples, $O(\frac{1}{\epsilon^2}(t(\log|\mathcal{F}| + \log t + c\log(B+c)) + \log\frac{1}{\delta}))$ samples drawn from $\mathcal{D}$ are sufficient to ensure that with probability at least $1 - \delta$ over the draw of the samples, the parameters $\hat{\alpha}, \hat{\beta}$ learned by ERM over the sample have expected loss that is at most $\epsilon$ larger than the expected loss of the best parameters $\alpha^*, \beta^* = \arg\min_{\alpha>0,\beta\geq1}\mathbb{E}_{D\sim\mathcal{D}}L(T_{\mathcal{F},(\hat{\alpha},\hat{\beta}),t}, D)$ over $\mathcal{D}$. Here $t$ is the size of the decision tree, $\mathcal{F}$ is the node function class used to label the nodes of the decision tree and $c$ is the number of label classes.*

*Proof of Theorem 3.3.* Since the loss is completely determined by the final decision tree $T_{\mathcal{F},(\alpha,\beta),t}$, it suffices to bound the number of different algorithm behaviors as one varies the hyperparameters $\alpha, \beta$ in Algorithm 1. As the tree is grown according to the top-down algorithm, suppose the number of internal nodes is $\tau < t$. There are $\tau + 1$ candidate leaf nodes to split and $|\mathcal{F}|$ candidate node functions, for a total of $(\tau + 1)|\mathcal{F}|$ choices for $(l, f)$. For any of $\binom{(\tau+1)|\mathcal{F}|}{2}$ pair of candidates $(l_1, f_1)$ and $(l_2, f_2)$, the preference for which candidate is 'best' and selected for splitting next is governed by the splitting functions $G_{\alpha,\beta}(T_{l_1\to f_1})$ and $G_{\alpha,\beta}(T_{l_2\to f_2})$. This preference flips across boundary condition given by $\sum_{l\in\text{leaves}(T_{l_1\to f_1})} w(l)g_{\alpha,\beta}(\{p_i(l)\}) = \sum_{l\in\text{leaves}(T_{l_2\to f_2})} w(l)g_{\alpha,\beta}(\{p_i(l)\})$. Most terms (all but three) cancel out on both sides as we substitute a single leaf node by an internal node on both LHS and RHS. The only unbalanced terms correspond to deleted leaves $l_1, l_2$ and newly introduced leaves $l_1^a, l_1^b, l_2^a, l_2^b$, i.e.

$$\sum_{l\in\{l_1^a,l_1^b,l_2\}} w(l)g_{\alpha,\beta}(\{p_i(l)\}) = \sum_{l\in\{l_2^a,l_2^b,l_1\}} w(l)g_{\alpha,\beta}(\{p_i(l)\}),$$

where $g_{\alpha,\beta}(\cdot) = g_{\alpha,\beta}^{\text{TSALLIS}}(\cdot)$, the $(\alpha, \beta)$-Tsallis entropy. Note that here $w(l)$ is a constant for a fixed problem instance (independent of the parameters $\alpha, \beta$ given the structure of the tree). For integer $\beta$, by the multinomial theorem, $(\sum_{i=1}^{c} p_i(l)^{\alpha})^{\beta}$ consists of at most $\binom{\beta+c-1}{c}$ distinct terms. By Rolle's theorem (more precisely, Lemma B.1), the number of distinct solutions of the above equation in $\alpha$ is $O((\beta + c)^c)$. Thus, for any fixed $\beta$ and fixed partial decision tree built in $\tau$ rounds, the number of critical points of $\alpha$ at which the $\arg\max$ in Line 3 of Algorithm 1 changes is at most $O(|\mathcal{F}|^2\tau^2(\beta + c)^c)$ and a fixed leaf node is split and labeled by a fixed $f$ for any interval of $\alpha$ induced by these critical points. Using a simple inductive argument over the number of rounds $t$ of Algorithm 1, this corresponds to at most $O(\Pi_{\tau=1}^{t}|\mathcal{F}|^2\tau^2(\beta + c)^c)$ critical points

across which the algorithmic behaviour (sequence of choices of node splits in Algorithm 1) can change as $\alpha$ is varied for a fixed $\beta$. Adding up over $\beta \in [B]$, we get $O(\sum_{\beta=1}^{B} |\mathcal{F}|^{2t} t^{2t} (\beta + c)^{ct})$, or at most $O(B|\mathcal{F}|^{2t} t^{2t} (B + c)^{ct})$ critical points.

This implies a bound of $O(t(\log |\mathcal{F}| + \log t + c \log(B + c)))$ on the pseudodimension of the loss function class by using Lemma A.2. Finally, an application of Theorem A.1 completes the proof. $\qquad\square$

**Theorem 3.4 (restated)** *Suppose $\gamma \in (0, 1]$. For any $\epsilon, \delta > 0$ and any distribution $\mathcal{D}$ over problem instances with $n$ examples, $O(\frac{1}{\epsilon^2}(t(\log |\mathcal{F}| + \log t) + \log \frac{1}{\delta}))$ samples drawn from $\mathcal{D}$ are sufficient to ensure that with probability at least $1 - \delta$ over the draw of the samples, the parameter $\hat{\gamma}$ learned by ERM over the sample is $\epsilon$-optimal, i.e. has expected loss at most $\epsilon$ larger than that of the optimal parameter over $\mathcal{D}$.*

*Proof of Theorem 3.4.* The loss is completely determined by the final decision tree $T_{\mathcal{F},\gamma,t}$. It suffices to bound the number of different algorithm behaviors as one varies the hyperparameter $\gamma$ in Algorithm 1. As the tree is grown according to the top-down algorithm, suppose the number of internal nodes is $\tau < t$. There are $\tau + 1$ candidate leaf nodes to split and $|\mathcal{F}|$ candidate node functions, for a total of $(\tau + 1)|\mathcal{F}|$ choices for $(l, f)$. For any of $\binom{(\tau+1)|\mathcal{F}|}{2}$ pair of candidates $(l_1, f_1)$ and $(l_2, f_2)$, the preference for which candidate is 'best' and selected for splitting next is governed by the splitting functions $G_\gamma(T_{l_1 \to f_1})$ and $G_\gamma(T_{l_2 \to f_2})$. This preference flips across boundary condition given by $\sum_{l \in \text{leaves}(T_{l_1 \to f_1})} w(l) g_\gamma(\{p_i(l)\}) = \sum_{l \in \text{leaves}(T_{l_2 \to f_2})} w(l) g_\gamma(\{p_i(l)\})$. Most terms (all but three) cancel out on both sides as we substitute a single leaf node by an internal node on both LHS and RHS. The only unbalanced terms correspond to deleted leaves $l_1, l_2$ and newly introduced leaves $l_1^a, l_1^b, l_2^a, l_2^b$, i.e.

$$\sum_{l \in \{l_1^a, l_1^b, l_2\}} w(l) g_\gamma(\{p_i(l)\}) = \sum_{l \in \{l_2^a, l_2^b, l_1\}} w(l) g_\gamma(\{p_i(l)\}).$$

Recall $g_\gamma(\{p_i\}) := C (\Pi_i p_i)^\gamma$, which implies that the above equation has six (i.e. $O(1)$) terms. By Rolle's theorem, the number of distinct solutions of the above equation in $\gamma$ is $O(1)$. Thus, the number of critical points of $\gamma$ at which the $\text{argmax}$ in Line 3 of Algorithm 1 changes is at most $O(|\mathcal{F}|^2 \tau^2)$ and a fixed leaf node is split and labeled by a fixed $f$ for any interval of $\gamma$ induced by these critical points. Over $t$ rounds, this corresponds to at most $O(\Pi_{\tau=1}^{t} |\mathcal{F}|^2 \tau^2) = O(|\mathcal{F}|^{2t} t^{2t})$ critical points across which the algorithmic behaviour (sequence of choices of node splits in Algorithm 1) can change as $\gamma$ is varied. This implies a bound of $O(t(\log |\mathcal{F}| + \log t))$ on the pseudodimension of the loss function class using Lemma A.2. An application of Theorem A.1 completes the proof. $\qquad\square$

**Theorem 3.5 (restated)** *Suppose $\sigma, \phi > 0$. For any $\epsilon, \delta > 0$ and any distribution $\mathcal{D}$ over problem instances with $n$ examples, $O(\frac{1}{\epsilon^2}(\log t + \log \frac{1}{\delta}))$ samples drawn from $\mathcal{D}$ are sufficient to ensure that with probability at least $1 - \delta$ over the draw of the samples, the parameters $\hat{\sigma}, \hat{\phi}$ learned by ERM over the sample have expected loss that is at most $\epsilon$ larger than the expected loss of the best parameters. Here $t$ denotes an upper bound on the size of the decision tree.*

*Proof of Theorem 3.5.* Fix the dataset $D$ and fix the random coins $\mathbf{z}$ used to generate the initial tree $T_{\mathbf{z},\sigma,\phi}$. We will use the piecewise loss structure to bound the Rademacher complexity, which would imply uniform convergence guarantees by applying standard learning-theoretic results.

First, we establish a piecewise structure of the dual class loss for fixed prior randomization $\mathbf{z}'$, $\ell_{\mathbf{z}}^D(\sigma, \phi) = \mathbb{E}_{\mathbf{z}'} L(T(T_{\mathbf{z},\sigma,\phi}, \mathbf{z}', \omega), D)$. Notice that the expected value under the remaining randomization $\mathbf{z}'$ is fixed, once the generated tree $T_{\mathbf{z},\sigma,\phi}$ is fixed. We first give a bound on the number of pieces of distinct trees generated as $\sigma, \phi$ are varied. The decision whether a node $\tau_i$ is split is governed by whether $p_{\text{SPLIT}}(\tau) = \sigma(1 + d_{\tau_i})^{-\phi} > \mathbf{z}_i$. Thus, we get at most $t - 1$ 2D curves in $\sigma, \phi$ across which the splitting decision may change. The curves are clearly monotonic. We further show that any pair of curves intersect in at most one point. Indeed, if $\sigma(1 + d_{\tau_i})^{-\phi} = \mathbf{z}_i$ and $\sigma(1 + d_{\tau_j})^{-\phi} = \mathbf{z}_j$, then $\phi' = \log(\mathbf{z}_j/\mathbf{z}_i)/\log\left(\frac{1+d_{\tau_i}}{1+d_{\tau_j}}\right)$ and $\sigma' = \mathbf{z}_i(1 + d_{\tau_i})^{\phi'}$ is the unique point provided $\phi' > 0$. Thus the set of all curves intersects in at most $\binom{t-1}{2} < t^2$ points. Since the curves are planar, the number of pieces in the dual loss function (or the number of distinct trees) is also $O(t^2)$. The above argument easily extends to a collection of $N$ problem instances, with a total of at most $O(t^2 N^2)$ pieces where distinct trees are generated across the instances.

Let $\rho_1, \ldots, \rho_m$ denote a collection of parameter values, with one parameter from each of the $m = O(N^2 t^2)$ pieces induced by all the dual class functions $\ell_{\mathbf{z}_i}^{D_i}(\cdot)$ for $i \in [N]$, i.e. across problems in the sample $\{D_1, \ldots, D_N\}$ for some fixed randomizations. Let $\mathcal{H} = \{f_\rho : (D, \mathbf{z}) \mapsto l_{\mathbf{z}}^D(\rho) \mid \rho \in \mathbb{R}^+ \times \mathbb{R}^+\}$ be a family of functions on a given sample of instances $S = \{D_i, \mathbf{z}_i\}_{i=1}^N$. Since the function $f_\rho$ is constant on each of the $m$ pieces, we have the empirical Rademacher complexity,

$$\hat{R}(\mathcal{H}, S) := \frac{1}{N}\mathbb{E}_\sigma\left[\sup_{f_\rho \in \mathcal{H}} \sum_{i=1}^N \sigma_i f_\rho(D_i, \mathbf{z}_i)\right]$$

$$= \frac{1}{N}\mathbb{E}_\sigma\left[\sup_{j \in [m]} \sum_{i=1}^N \sigma_i f_{\rho_j}(D_i, \mathbf{z}_i)\right]$$

$$= \frac{1}{N}\mathbb{E}_\sigma\left[\sup_{j \in [m]} \sum_{i=1}^N \sigma_i v_{ij}\right],$$

where $\sigma = (\sigma_1, \ldots, \sigma_m)$ is a tuple of i.i.d. Rademacher random variables, and $v_{ij} := f_{\rho_j}(D_i, \mathbf{z}_i)$. Note that $v^{(j)} := (v_{1j}, \ldots, v_{Nj}) \in [0, H]^N$, and therefore $\|v^{(j)}\|_2 \le H\sqrt{N}$, for all $j \in [m]$. An application of Massart's lemma Massart [2000] gives

$$\hat{R}(\mathcal{H}, S) = \frac{1}{N}\mathbb{E}_\sigma\left[\sup_{j \in [m]} \sum_{i=1}^N \sigma_i v_{ij}\right]$$

$$\le H\sqrt{\frac{2\log m}{N}}$$

$$\le H\sqrt{\frac{4\log Nt}{N}}.$$

Standard Rademacher complexity bounds [Barlett et al. 2002] now imply the desired sample complexity bound.

$\square$

**Theorem 3.6 (restated)** *Suppose $p \in [0, 1]$. For any $\epsilon, \delta > 0$ and any distribution $\mathcal{D}$ over problem instances with $n$ examples, $O(\frac{1}{\epsilon^2}(t(\log|\mathcal{F}| + n) + \log\frac{1}{\delta}))$ samples drawn from $\mathcal{D}$ are sufficient to ensure that with probability at least $1 - \delta$ over the draw of the samples, the Tweedle power parameter $\hat{p}$ learned by ERM over the sample is $\epsilon$-optimal.*

*Proof (of Theorem 3.6).* The loss is completely determined by the final decision tree $T_{\mathcal{F},p,t}$. It suffices to bound the number of different algorithm behaviors as one varies the hyperparameter $p$ in Algorithm 1. As the tree is grown according to the top-down algorithm, suppose the number of internal nodes is $\tau < t$. For any of $\binom{(\tau+1)|\mathcal{F}|}{2}$ pair of candidates $(l_1, f_1)$ and $(l_2, f_2)$, the preference for which candidate is 'best' and selected for splitting next is governed by the splitting functions $G_p(T_{l_1 \to f_1})$ and $G_p(T_{l_2 \to f_2})$. This preference flips across boundary condition given by $\sum_{l \in \text{leaves}(T_{l_1 \to f_1})} w(l)g_p(\{p_i(l)\}) = \sum_{l \in \text{leaves}(T_{l_2 \to f_2})} w(l)g_p(\{p_i(l)\})$. The expression simplifies and the only remaining terms correspond to deleted leaves $l_1, l_2$ and newly introduced leaves $l_1^a, l_1^b, l_2^a, l_2^b$, i.e. $\sum_{l \in \{l_1^a, l_1^b, l_2\}} w(l)g_p(\{p_i(l)\}) = \sum_{l \in \{l_2^a, l_2^b, l_1\}} w(l)g_p(\{p_i(l)\})$.

Recall $g_p(\{p_i\})$ gives an equation in $O(|y_l|) = O(n)$ terms. By Rolle's theorem, the number of distinct solutions of the above equation in $p$ is $O(n)$. Thus, the number of critical points of $p$ at which the $\mathrm{argmax}$ in Line 3 of Algorithm 1 changes is at most $O(|\mathcal{F}|^2\tau^2 n)$ and a fixed leaf node is split and labeled by a fixed $f$ for any interval of $p$ induced by these critical points. Over $t$ rounds, this corresponds to at most $O(\Pi_{\tau=1}^t |\mathcal{F}|^2\tau^2 n) = O(|\mathcal{F}|^{2t}t^{2t}n^t)$ critical points across which the algorithmic behaviour (sequence of choices of node splits in Algorithm 1) can change as $p$ is varied. This implies a bound of $O(t(\log|\mathcal{F}| + \log t + n)) = O((\log|\mathcal{F}| + n))$ on the pseudodimension of the loss function class using Lemma A.2, since $t \le n$. An application of Theorem A.1 completes the proof. $\square$

## C  PROOFS FROM SECTION 4

**Theorem 4.1 (restated)** *Suppose $\tilde{\alpha} \in \mathbb{R}_{\ge 0}$ and $t$ denote the size of the unpruned tree. For any $\epsilon, \delta > 0$ and any distribution $\mathcal{D}$ over problem instances with $n$ examples, $O(\frac{1}{\epsilon^2}(\log t + \log\frac{1}{\delta}))$ samples drawn from $\mathcal{D}$ are sufficient to ensure that with probability at least $1 - \delta$ over the draw of the samples, the mininum cost-complexity pruning parameter learned by ERM over the sample is $\epsilon$-optimal.*

*Proof of Theorem 4.1.* Fix a dataset $D$. Then there are critical values of $\tilde{\alpha}$ given by $\tilde{\alpha}_0 = 0 < \tilde{\alpha}_1 < \tilde{\alpha}_2 \cdots < \infty$ such that the optimal pruned tree $T_k$ is fixed for over any interval $[\tilde{\alpha}_k, \tilde{\alpha}_{k+1})$ for $k \geq 0$. Furthermore, the optimal pruned trees form a sequence of nested sub-trees $T_0 = T \supset T_1 \supset \ldots$ (Breiman et al. [1984], Chapter 10). Thus, the behavior of the min cost-complexity pruning algorithm is identical over at most $t$ intervals, and the loss function is piecewise constant with at most $t$ pieces. The rest of the argument is similar to the proof of Theorem 3.3, and we obtain a pseudo-dimension bound of $O(\log t)$ using Lemma A.2. An application of Theorem A.1 implies the stated sample complexity. $\square$

**Theorem 4.2 (restated)** *Suppose $c_1, c_2 \in \mathbb{R}_{\geq 0}$ and $t$ denote the size of the unpruned tree. For any $\epsilon, \delta > 0$ and any distribution $\mathcal{D}$ over problem instances with $n$ examples, $O(\frac{1}{\epsilon^2}(\log t + \log \frac{1}{\delta}))$ samples drawn from $\mathcal{D}$ are sufficient to ensure that with probability at least $1 - \delta$ over the draw of the samples, the pessimistic pruning parameters learned by ERM over the sample is $\epsilon$-optimal.*

*Proof of Theorem 4.2.* For a fixed dataset $D$, the $c_1, c_2$ parameter space can be partitioned by at most $t$ algebraic curves of degree 3 that determine the result of the pessimistic pruning test. We use a general result on the pseudodimension bound in data-driven algorithm design due to Bartlett et al. [2022] when the loss can be computed by evaluating rational expressions to obtain a $O(\log t)$ on the pseudodimension. The result is stated below for convenience.

In this theorem, our above arguments show that there is a GJ algorithm, i.e. an algorithm which only computes and compares rational (ratios of polynomials) functions of its inputs, for computing the loss function. Here the number of real parameters $n = 2$, the maximum degree of any computed expression is $\Delta = 3$ and the total number of distinct predicates that need to be evaluated to compute the loss for any value of the parameters is $\Gamma = t$. Plugging into Theorem A.3 yields a bound of $O(\log t)$ on the pseudo-dimension, and the result follows from Theorem A.1. $\square$

# D  PROOFS FROM SECTION 5

**Theorem 5.1 (restated)** *Suppose $\alpha > 0, \beta \in [B], \tilde{\alpha} \geq 0$. For any $\epsilon, \delta > 0$ and any distribution $\mathcal{D}$ over problem instances with $n$ examples, $O(\frac{1}{\epsilon^2}(t(\log |\mathcal{F}| + \log t + c\log(B + c)) + \log \frac{1}{\delta}))$ samples drawn from $\mathcal{D}$ are sufficient to ensure that with probability at least $1 - \delta$ over the draw of the samples, the parameters learned by ERM for $L_\eta$ are $\epsilon$-optimal.*

*Proof of Theorem 5.1.* As argued in the proof of Theorems 3.3, there is a bound of $O(B|\mathcal{F}|^{2t}t^{2t}(B+c)^{ct})$ on the number of distinct algorithmic behavior of the top-down learning algorithm in growing a tree of size $t$ as the parameters $\alpha, \beta$ are varied. Further, as argued in the proof of Theorem 4.1, for each of these learned trees, there are at most $t$ distinct pruned trees as $\tilde{\alpha}$ is varied. Overall, this corresponds to $O(B|\mathcal{F}|^{2t}t^{2t+1}(B+c)^{ct})$ distinct behaviors, which implies the claimed sample complexity bound using standard tools from learning theory and data-driven algorithm design (Lemma A.2, Theorem A.1). $\square$

# E  ADDITIONAL EXPERIMENTS

We include further experiments below for the interested reader. In the following we observe that the explainability-accuracy frontier depends on the splitting criterion, and further examine the tuning of $(\alpha, \beta)$-Tsallis entropy on additional datasets.

## E.1  EXPLAINABILITY-ACCURACY FRONTIER

We study the effect of varying $\alpha$ (for fixed $\beta = 1$) and $\beta$ (for fixed $\alpha = 1.5$) on the explainability-accuracy trade-off. We fix $\eta = 0.01$, and obtain the plot by varying the amount of pruning by changing the complexity parameter $\tilde{\alpha}$ in min-cost complexity pruning.

We perform this study for Iris and Wine datasets in Figure 4. We observe that for a given accuracy, the best (smallest) explanation (size) could be obtained for different different splitting criteria (corresponding to setting of $\alpha, \beta$). In particular, different criteria can dominate in different regimes of size and $\eta$. Therefore, simultaneously tuning splitting criterion and pruning as in Theorem 5.1 is well-motivated.

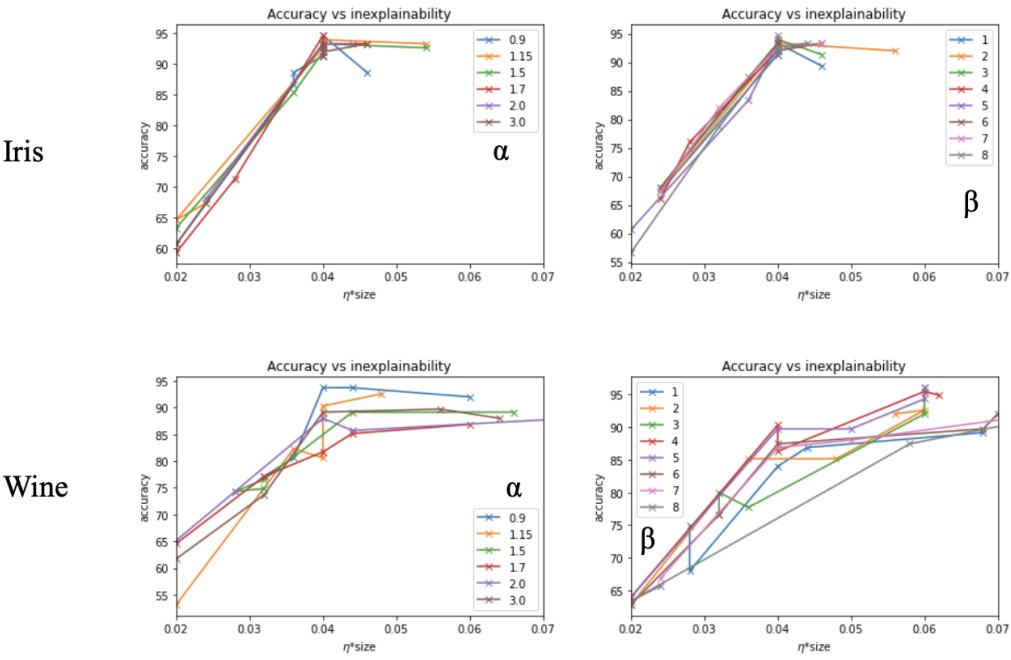

Figure 4: Accuracy-explainability frontier for different $\alpha$ or different $\beta$, as the pruning parameter $\tilde{\alpha}$ is varied.

## E.2 $(\alpha, \beta)$-TSALLIS ENTROPY

We consider several additional datasets from the UCI repository and examine the best setting of $(\alpha, \beta)$ in the splitting criterion. The results are depicted in Figure 5 and summarized below.

Seeds Charytanowicz et al. [2012] involves 3 classes of wheat, and has 210 instances with 7 attributes each. The splitting criterion proposed by Kearns and Mansour [1996] seems to work best here. Note that the original work only studied binary classification, and seeds involves three label classes and therefore our experiment involves a natural generalization of Kearns and Mansour [1996] to $g_{\frac{1}{2},2}^{\text{TSALLIS}}(\cdot)$.

Cryotherapy Khozeimeh et al. [2018] has 90 instances with 7 real or integral attributes and contains the binary label of whether a wart was suffessfully treated using cryotherapy. Here $\alpha = 0.5$ with $\beta = 4$ is one of the best settings, indicating usefulness of varying the $\beta$ exponent in the KM96 criterion.

Glass identification German [1987] involves classification into six types of glass defined in terms of their oxide content. There are 214 instances with 9 real-valued features. Interestingly, the best performance is observed when both $\alpha$ and $\beta$ are larger than their typical values in popular criteria. For example, $(\alpha, \beta) = 2.45, 6$ works well here.

Algerian forest fires involves binary classification with 12 attributes and 244 instances. Gini entropy by itself does poorly, but augmented with the $\beta$-parameter the performance improves significantly and beats other candidate approaches for $\beta = 8$.

Human activity detection using smartphones Reyes-Ortiz et al. [2012] is a 6-way classification dataset consisting of smartphone accelerometer and gyroscope readings corresponding to different activities, with 10299 instances with 561 features. Smaller values of $\alpha$ work better on this dataset, and the dependence on $\beta$ is weaker.

## E.3 PRUNING EXPERIMENTS

We will examine the effectiveness of learning to prune by comparing the accuracy of pruning using the learned parameter $\tilde{\alpha}$ in the mininum cost-complexity pruning algorithm family with other baseline methods studied in the literature. Prior literature on empirical studies on pruning methods has shown that different pruning methods can work best for different datasets Mingers [1989a], Esposito et al. [1997]. This indicates that a practitioner should try out several pruning methods in order to obtain the best result for given domain-specific data. Here we will show that a well-tuned pruning from a single

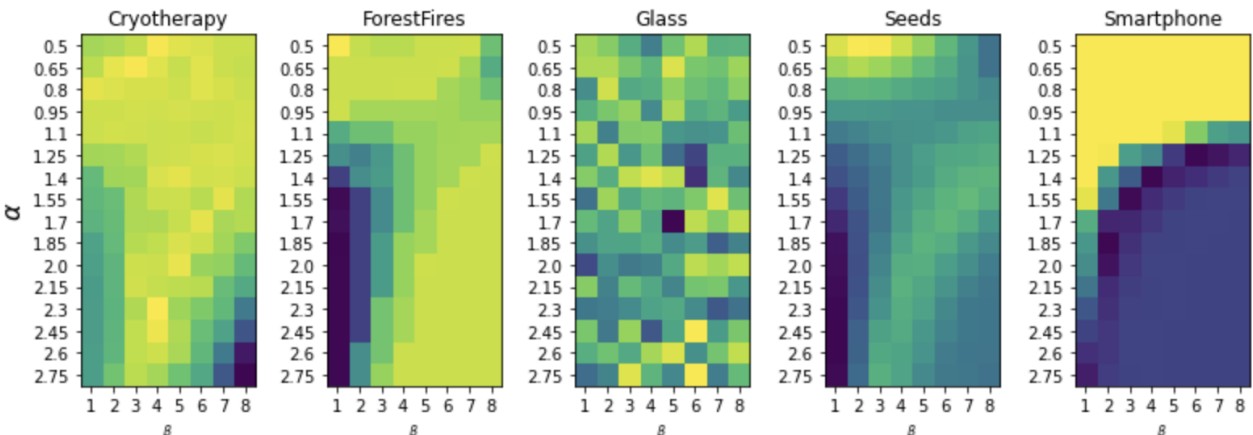

Figure 5: Average test accuracy (proportional to brightness) of $(\alpha, \beta)$-Tsallis entropy based splitting criterion across additional datasets.

| Dataset | Acc(Unpruned) | Acc($\tilde{\alpha}^*$) in MCCP | Acc(REP) | Acc(TDP) | Acc(BUP) |
|---|---|---|---|---|---|
| Iris | 80.03 | 97.37 | 96.67 | 90.00 | 93.33 |
| Digits | 84.44 | 89.42 | 86.67 | 83.61 | 88.89 |
| Breast cancer | 87.72 | 93.71 | 92.98 | 91.23 | 92.11 |
| Wine | 80.56 | 94.44 | 91.67 | 88.89 | 86.11 |

Table 2: A comparison of the mean test accuracy of decision trees obtained using different pruning methods.

algorithm family can be competitive, and allows us to automate this process of manual selection of the pruning algorithm.

We perform our experiments on benchmark datasets from the UCI repository, including Iris, Wine, Breast Cancer and Digits datasets. We split the datasets into train-test sets, using 80% instances for training and 20% for testing. In each case, we build the tree using entropy as the splitting criterion. We compare the mean accuracy on the test sets over 50 different splits for the following methods:

- Unpruned, that is no pruning method is used.
- $\tilde{\alpha}^*$ in MCCP. Min-cost complexity pruning using the best parameter $\tilde{\alpha}^*$ for the dataset.
- REP, Reduced error pruning method of Quinlan [1987].
- TDP, Top-down pessimistic pruning method of Quinlan [1986].
- BUP, Bottom-up pessimistic pruning method of Mansour [1997].

We report our findings in Table 2. We observe that the learned pruning method has a better mean test accuracy than other baseline methods on the tested datasets.