# OpenReview forum: "Learning Accurate and Interpretable Decision Trees"
_auai.org/UAI/2024/Conference — UAI 2024 oral_

### Official Review · Reviewer_T6Jg · 2024-03-12

**Q2-1 Originality-Novelty:** 4
**Q2-2 Correctness-Technical Quality:** 4
**Q2-5 Clarity Of Writing:** 4

**Q1 Summary And Contributions:**

The paper tackles the problem of learning decision trees, a machine learning model that is re-gaining come interest due to its' simplicity, efficiency and also the capability to be "explained" or at least "given an interpretation".
The main contributions of the paper are the following:
1) a novel family of node splitting methods are proposed, based on the Tsallis criterion
2) study of how to perform parameter tuning in Bayesian decision tree learning algorithms for generating the prior distribution.
3) a parameterized family for node splitting, in the case of regression trees, is studied together with sample complexity bounds for the tuning parameter.
4) explainability - accuracy trade-off is optimised to design decision tree learning algorithms
5) show how to tune parameters for most used decision tree algorithms including when including the complexity parameter

**Q2-3 Extent To Which Claims Are Supported By Evidence:**

4: Excellent: all claims are supported by very convincing evidence (in the form of comprehensive experimental evaluation, rigorous mathematical proofs, detailed (pseudo-)code, precise references, well-motivated and realistic assumptions) and the authors deliver what they promise.

**Q2-4 Reproducibility:**

2: Fair: key resources (e.g. proofs, code, data) are unavailable but key details (e.g. proof sketches, experimental setup) are sufficiently well-described for an expert to confidently reproduce the main results.

**Q3 Main Strengths:**

1) the tackled problem is relevant
2) the structure and organization of the paper is optimal
3) several contributions are made and are well supported in theoretical terms
4) the paper gives theoretical bounds for practica learning of decision trees

**Q4 Main Weakness:**

1) numerical experiments should be extended
2) numerical experiments should take into account datasets with more attributes than those of the considered datasets
3) explainability has not been analyzed in a satisfying manner, it could be more detailed and articulated than it is as of now

**Q5 Detailed Comments To The Authors:**

I enjoyed reading the paper which is about a relevant problem and gives useful contributions both in terms of theory and practice to the research community of machine learning. I think that your very nice work could benefit from more numerical experiments, with a larger number of attributes, by comparing also with other machine learning algorithms and not only with other decision trees algorithms. I think that the comparison you did is fair enough but in he end when one selects the appropriate machine learning model would like to know how attractive is a decision tree learnt thanks to your improvements. I also would like to know more about the issue of explainability, which seems to be something where you can make some useful contribution, but I fear that this aspect could be overlooked in the current form, because of no comparison to other approaches.

**Q9 Complying With Reviewing Instructions:**

Yes

---

> ### Author Rebuttal · Authors · 2024-04-06
>
> We thank the reviewer for their time and for providing useful feedback.
>
>
> **Potential for further experiments**. We appreciate the suggestion for including datasets with more attributes in our experiments. We also plan to include more baselines for pruning in our experiments, and believe there is scope for interesting extensions to our work by applying our approaches to tuning other methods of pruning used in the literature.
>
>
>
>
> **On explainability**. We find this relatively more challenging to compare the explainability of decision trees with other machine learning models due to the lack of a standard definition of explainability. In the context of decision trees, prior literature recommends that the size of the tree is a good measure of its explainability (smaller trees are easier to interpret/understand). Decision trees naturally “explain” the features involved in making any prediction, and in other models like deep neural networks additional steps are needed to make the model more explainable. In the context of clustering, the explainability notion is connected to how well the clustering can be mapped to a decision tree of small size [1, 2].
>
> *References*
>
> [1] Moshkovitz, Michal, Sanjoy Dasgupta, Cyrus Rashtchian, and Nave Frost. "Explainable k-means and k-medians clustering." In International conference on machine learning, pp. 7055-7065. PMLR, 2020.
>
> [2] Gupta, Anupam, Madhusudhan Reddy Pittu, Ola Svensson, and Rachel Yuan. "The price of explainability for clustering." In 2023 IEEE 64th Annual Symposium on Foundations of Computer Science (FOCS), pp. 1131-1148. IEEE, 2023.

---

### Official Review · Reviewer_MSZu · 2024-03-14

**Q2-1 Originality-Novelty:** 2
**Q2-2 Correctness-Technical Quality:** 3
**Q2-5 Clarity Of Writing:** 2

**Q1 Summary And Contributions:**

The submission addresses learning decision or regression trees using empirical risk minimization. More precisely, the paper considers using a parameterized family of loss functions (or splitting criteria), namely $(\alpha,\beta)$-Tsallis entropy for classification and mean Tweedle deviance error with power $p$ for regression. The paper studies theoretical bounds on the expected loss obtained by learning the parameters of these splitting criteria using ERM. The paper further addresses the problem of pruning the tree, which can be used to learn trees with a good cost-complexity tradeoff.

**Q2-3 Extent To Which Claims Are Supported By Evidence:**

3: Good: the main claims are supported by convincing evidence (in the form of adequate experimental evaluation, proofs, (pseudo-)code, references, assumptions).

**Q2-4 Reproducibility:**

3: Good: key resources (e.g. proofs, code, data) are available and key details (e.g. proofs, experimental setup) are sufficiently well-described for competent researchers to confidently reproduce the main results.

**Q3 Main Strengths:**

The paper addresses an interesting problem; its impact is reasonable, decision trees being still broadly used.

The paper features several theorems supporting the claims made.

**Q4 Main Weakness:**

The actual gain in terms of generalization power remains difficult to appreciate; the experimental results displayed do not show a significant improvement over the other (classical) splitting criteria, on the four datasets considered in the experiments. In addition, how the proposal compares to tree ensembles (such as, e.g., random forests), which remain the most frequently used approach involving trees, is not clear. How the results presented here could be extended to tree ensembles is not obvious, also due to the computational complexity of the approach being unclear.

The overall complexity of the proposed approach is not clear, and the paper hints that optimizing over the parameterized family of splitting criteria may have a substantial computational cost.

The paper does not address the issue of the explainability of the model, as suggested by the title—rather, the theorems exposed can be used to determine to which extent the tree can be pruned without losing too much generalization power.

The paper is difficult to follow, due to its heavy content. In my opinion, the writing should be improved by a significant margin.

**Q5 Detailed Comments To The Authors:**

The paper is interesting and addresses a meaningful problem, trees being still used across the scientific community and in many industrial applications. Nevertheless, I have the following questions and comments.

First, it remains difficult to appreciate to which extent the proposal makes it possible to improve the performances of the model. In practice, the differences in terms of performance are not significant: although the accuracies displayed are consistently higher for the proposal, the difference remains small, and within reach in terms of confidence intervals.

Can you provide any insights on these differences, for instance over other datasets ? Can you as well shed some light on the overall complexity of the learning procedure proposed ?

Although trees remain used, they are widely employed as base algorithm for ensembles of classifiers, such as random forests. Can you provide some insights regarding how the results presented in the current paper could be extended to tree ensembles ?

The issue of explainability is not really covered in the paper. Although the proposal can be used to guarantee that the pruning will not degrade (too much) the performances, explainability is not explicitly taken into account in the training. Can you elaborate on that ?

The writing should be improved—by a substantial amount. It is too dense. Some of the basic material is not recalled—such as, e.g., the paper by Kearns and Mansour (1996). Some notions should be explicitly stated (e.g., permissibility), page 4). Theorem 3.3, for instance, is hard to understand, as well as its proof sketch. The "appropriate choice of $C$" page 5 should be made clear.

The equations should be numbered.

The paper should be checked for typos, a number of which can be found (e.g., "multile" page 1, "a framework for design" page 2, "class of splitting criterion" page 3, math $\beta=1$ at the beginning of a sentence page 4, $Q=\{p_{\pi(i)}$ missing a bracket in Proposition 3.2, "elastic net" having several spellings, "lloyd" missing a capital in the references, and many more).

**Q9 Complying With Reviewing Instructions:**

Yes

---

> ### Author Rebuttal · Authors · 2024-04-06
>
> We thank the reviewer for their time and for raising some interesting questions.
>
>
> **Improvement in performance**. We have reported 95% confidence intervals at which selecting the best hyperparameter already beats the popular Gini impurity criterion on multiple datasets (including Banknote, Wine). Our mean performance is better than other alternatives as well, but with a smaller confidence level i.e. on fewer than 95% but still the majority of data samples. We can include this measure (fraction of data samples where the learned criterion beats each baseline criterion) as a clearer indicator of improved performance.
>
>
> **Overall complexity of learning**. We provide polynomial sample complexity bounds for learning the best parameters for various algorithm families used for building decision trees. These learning guarantees correspond to the ERM (Empirical Risk Minimization) algorithm, which computes the parameter which minimizes the sample loss (e.g. Pg 4, top right). Computationally efficient implementation of the ERM is an interesting research question, but we remark that sample efficiency is an important problem in itself and several works in the data-driven algorithm design have exclusively focused on the sample complexity of learning (e.g. [1, 2]).
>
>
> **Extension to tree ensembles**. Extension of our approaches to tree ensembles is an interesting question, although this comes at the expense of making the model less interpretable. We still need to choose splitting and pruning methods used in building the individual trees. If we learn a uniform splitting criterion for all trees, our sample complexity arguments are straightforward to extend to this case and would imply an additional $O(n_t)$ factor in the sample complexity, where $n_t$ is the number of trees in the random forest (in the case of pruning, our arguments would imply an $O(\log n_t)$ term). There are interesting further questions here, including learning a combination of splitting/pruning criteria across different trees and tuning the number of trees $n_t$ as a hyperparameter (which impacts both accuracy and interpretability). We are happy to add a discussion about this in the final version.
>
> **Explainability**. There is a subtle difference between pruning and incorporating explainability in the loss objective itself as we explain in Section 5. While pruning using the min cost-complexity approach, we only add the size of the tree as a regularization term to the training objective and still measure the accuracy of prediction on the test set (Section 4). In Section 5, we also incorporate the size of the tree in the loss objective for test data following recent works (Hu et al. 2019 and Lin et al. 2020), and study the sample complexity of simultaneously learning the splitting criterion and pruning hyperparameter.
>
>
>
>
> We thank the reviewer for pointing out the typos and minor issues and will carefully proofread and revise the paper to incorporate the suggestions to improve the readability of our draft.
>
>
> *References*
>
> [1] Avrim Blum, Chen Dan, and Saeed Seddighin. Learning complexity of simulated annealing. In International Con- ference on Artificial Intelligence and Statistics (AISTATS), pages 1540–1548. PMLR, 2021.
>
> [2] Peter Bartlett, Piotr Indyk, and Tal Wagner. Generalization bounds for data-driven numerical linear algebra. In Con- ference on Learning Theory, pages 2013–2040. PMLR, 2022.

---

### Official Review · Reviewer_TcSy · 2024-03-22

**Q2-1 Originality-Novelty:** 3
**Q2-2 Correctness-Technical Quality:** 3
**Q2-5 Clarity Of Writing:** 3

**Q1 Summary And Contributions:**

This paper proposes a novel splitting criterion for decision trees which interpolates popular known methods.
It works for classification as well as regression, and theoretical results confirm that this allows to select hyper parameters to improve accuracy.
Experimental results illustrate the method.

**Q2-3 Extent To Which Claims Are Supported By Evidence:**

3: Good: the main claims are supported by convincing evidence (in the form of adequate experimental evaluation, proofs, (pseudo-)code, references, assumptions).

**Q2-4 Reproducibility:**

4: Excellent: key resources (e.g. proofs, code, data) are available and key details (e.g. proof sketches, experimental setup) are comprehensively described for competent researchers to confidently and easily reproduce the main results.

**Q3 Main Strengths:**

- the new splitting criterion seems to be relevant, theoretically and experimentally
- i like the idea of considering an objective function that takes into account the complexity not only for overfitting, but here for interpretability, and the derived theorem with a neighbor loss.
- they look at BART, which is a nice bayesian extension of decision trees which has many hyperparameter to tune, so perfect fit for this paper

**Q4 Main Weakness:**

The structure of the paper may be improved
I am not sure how the different datasets are constructed in the real dataset, in practice.

**Q5 Detailed Comments To The Authors:**

I am not sure to understad the last sentence of paragraph 1 in Section 2
"We do not assume that individual data points (X_i,y_i) are iid in any dataset D_j"
Does this means that a dataset is a sample, but the observations within different datasets are not iid?

Figure 1 is not at the right place, in a paragraph of regression but dealing with classification

Typos:
last paragraph of P1, multile -> multiple
footnote 1, the our

**Q9 Complying With Reviewing Instructions:**

Yes

---

> ### Author Rebuttal · Authors · 2024-04-06
>
> We thank the reviewer for their time and for providing useful feedback. We provide clarifications below.
>
>
> **How different datasets are constructed in experiments**. For the purpose of experiments, we simulate different datasets from the same domain by taking random subsets of the same dataset. This corresponds to k-fold cross-validation for which our results imply convergence guarantees. In practice, this framework is useful when we have new data being regularly generated, or when we have access to several related datasets.
>
> **"We do not assume that individual data points $(X_i,y_i)$ are iid in any dataset $D_j$"**. We assume that there is a meta-distribution (over finite datasets with some maximum size $m$) which generates the dataset $D_j=X^{(j)},y^{(j)}$ (this is the sense in which different datasets are drawn from the same application domain) but individual datapoints $X^{(j)}_1,y^{(j)}_1, X^{(j)}_2,y^{(j)}_2, \dots, X^{(j)}_m,y^{(j)}_m$ within the datasets are not assumed to be iid.
>
>
>
> We will also take care to address other typos and suggestions noted by the reviewer.

---

### Official Review · Reviewer_rLWX · 2024-03-22

**Q2-1 Originality-Novelty:** 3
**Q2-2 Correctness-Technical Quality:** 3
**Q2-5 Clarity Of Writing:** 4

**Q10 Ethical Concerns:**

No.

**Q1 Summary And Contributions:**

The motivations and proposal of the paper are clearly articulated and so are the contributions. The authors introduced a new approach to enhance the trade-off between interpretability and accuracy, using induction mechanism of decision trees. They proposed a method for automatically designing a decision tree learning algorithms by selecting hyperparameters through data-driven techniques across multiple problem instances within the same domain. In this perspective, they investigate the minimum number of problem instances required to learn a reliably good algorithm (hyperparameter) under the statistical learning framework, considering various design perspectives crucial in decision tree construction. One of the key contributions is the introduction of a new family of node splitting criteria based on Tsallis entropy metric. Additionally, the authors address and enhance aspects of optimizing the interpretability-accuracy trade-off in designing decision tree learning algorithms. This includes simultaneously tuning splitting and pruning parameters during the growth of a decision tree to size "t" and its subsequent pruning down to size "t' ≤ t", while minimizing an objective function that incorporates both explainability and accuracy.

**Q2-3 Extent To Which Claims Are Supported By Evidence:**

3: Good: the main claims are supported by convincing evidence (in the form of adequate experimental evaluation, proofs, (pseudo-)code, references, assumptions).

**Q2-4 Reproducibility:**

3: Good: key resources (e.g. proofs, code, data) are available and key details (e.g. proofs, experimental setup) are sufficiently well-described for competent researchers to confidently reproduce the main results.

**Q3 Main Strengths:**

This paper addressed a relevant problem and introduced a significant novel contribution, providing important results to enhance the trade-off between interpretability and accuracy, using induction mechanism of decision trees.  The paper is quite readable, well written, and well-organized, as well as it is technically sound. The used evaluation methods, including the theoretical validation for the stated results, are appropriate for the intended contribution The research topic introduced in this paper is interesting and relevant to the UAI audience.

**Q4 Main Weakness:**

Overall, the paper is well written, but minor issues could be improved, for instance, Figure 2 is in Section 4, but it only will be explained in Section 6. The conclusion Section should be improved, providing more discussion on the main limitations in the obtained results, as well as some proposition for future work.  Apparently,  you used a statistical test, but it was not clear what it was. Therefore, I suggest informing which test was selected, the sample size, among other relevant points.

**Q5 Detailed Comments To The Authors:**

Overall, the paper is well written, but minor issues could be improved, for instance, Figure 2 is in Section 4, but it only will be explained in Section 6. The Conclusion Section should be improved, providing more discussion on the main limitations in the obtained results, as well as some proposition for future work.   Apparently, you used a statistical test, but it was not clear what it was. Therefore, I suggest informing which test was selected, the sample size, among other relevant points. Finally, It would be interesting to create an Anonymous GitHub (https://anonymous.4open.science/)  with the code, as it would make it more practical for future research on the subject.

**Q9 Complying With Reviewing Instructions:**

Yes

---

> ### Author Rebuttal · Authors · 2024-04-06
>
> We thank the reviewer for their time and for providing useful feedback.
>
>
> **Conclusion section**. We thank the reviewer for pointing out areas for improvement of the conclusion section by discussing limitations and future directions. We note that a limitation of our work is the unresolved challenge of computationally efficient optimization of the sample accuracy, our theoretical results focus on sample efficiency. We also remark that we focus on upper bounds on sample complexity, and do not provide corresponding lower bounds. Another direction for future research as noted by another reviewer is designing and analyzing a potentially more powerful algorithm family for pruning.
>
>
> **Suggestions for code**. We provide standard student’s $t$ distribution based confidence intervals for 95% confidence. We will include the details about this confidence interval computation, including the number of iterations used in our final version. We also thank the reviewer for suggesting uploading our code to a publicly available repository, which we agree will be helpful for future research.

---

### Official Review · Reviewer_Uekh · 2024-03-26

**Q2-1 Originality-Novelty:** 3
**Q2-2 Correctness-Technical Quality:** 3
**Q2-5 Clarity Of Writing:** 4

**Q1 Summary And Contributions:**

This paper tackles the intricate task of crafting decision tree learning algorithms by framing it as a hyperparameter selection challenge spanning multiple problem instances from the same domain. The paper introduces a family of node splitting criteria and delving into parameter tuning in Bayesian decision tree methodologies, it offers insights into sample complexity boundaries while navigating the delicate balance between explainability and accuracy. To sum up, this is a paper on data-driven decision tree design strategies.

**Q2-3 Extent To Which Claims Are Supported By Evidence:**

3: Good: the main claims are supported by convincing evidence (in the form of adequate experimental evaluation, proofs, (pseudo-)code, references, assumptions).

**Q2-4 Reproducibility:**

3: Good: key resources (e.g. proofs, code, data) are available and key details (e.g. proofs, experimental setup) are sufficiently well-described for competent researchers to confidently reproduce the main results.

**Q3 Main Strengths:**

I greatly appreciated the effort to generalize the splitting and pruning criteria and the ability to frame everything within a data-driven decision tree design strategy. Although I wasn't able to follow all the formal details, it seems to me that the theoretical foundation is solid. The experimental section is rather limited and serves primarily to demonstrate the feasibility of the approach.

**Q4 Main Weakness:**

In the proposed approach, I identify two issues. One concerns the referencing of background literature. Mingers' work on Decision Tree pruning has been criticized for several errors, as demonstrated in:

Floriana Esposito, Donato Malerba, Giovanni Semeraro: A Comparative Analysis of Methods for Pruning Decision Trees. IEEE Trans. Pattern Anal. Mach. Intell. 19(5): 476-491 (1997)

and further discussed in:

Esposito, F., Malerba, D., Semeraro, G., Tamma, V. Effects of pruning methods on the predictive accuracy of induced decision trees. Applied Stochastic Models in Business and Industry, 1999, 15(4), pp. 277–299.

The other issue is that with the great variety of pruning algorithms, the part on decision tree pruning is not developed in deep and the experimental section is very limited.

**Q5 Detailed Comments To The Authors:**

In the abstract, please revise the sentence "In this work, we design approaches to design decision tree learning algorithms given repeated access to data from the same domain." trying to avoid the use of the verb design twice.

multile --> multiple

Try to link the problem formulation (the learner has access to multiple related datasets D1, . . . ,DN coming from the same
problem domain) to transfer learning. Indeed, this scenario inherently aligns with the principles of transfer learning, where knowledge gained from one dataset can be leveraged to enhance learning performance on related datasets within the same domain. Surprisingly I didn't find any reference to transfer learning.

In Figure 3, I assume that the dark blue indicates a low test accuracy, while the light yellow represents higher test accuracy. This interpretation is based on the assumption that increasing brightness in the graph corresponds to an increase in test accuracy. This should be made explicit in the paper, since it is not clear. Having said this, how to interpret the case of the Iris data, there the light yellow is at two opposite corners of the map? Any comment on this?

**Q9 Complying With Reviewing Instructions:**

Yes

---

> ### Author Rebuttal · Authors · 2024-04-06
>
> We thank the reviewer for their time and for providing useful feedback.
>
>
> **Pruning literature**. We thank the reviewer for pointing out the more updated literature on empirical comparison of different methods for pruning decision trees. While our results achieve unification of major approaches for splitting nodes in top-down construction of the decision tree, allowing automatic selection of domain-specifc optimal splitting criterion, our pruning results show how to set tunable parameters in a couple of distinct popular methods including the min-cost complexity pruning. Given the vast variety of pruning methods in the literature (for example, the Esposito et al. paper you mention compares at least six distinct methods) we appreciate the comment that a deeper study that enables selection of domain-specific pruning methods would be very desirable. Based on your suggestion, we plan to incorporate the alternative pruning methods as baselines in our experiments and we will add that a deeper study for selecting the pruning method is an important future research direction.
>
>
>
>
> **Connection to transfer learning**. We appreciate the reviewer’s suggestion for connecting our work with transfer learning. We would like to note that some prior work [1, 2] has formalized the connection between the data-driven algorithm design framework which we adopt in our work, and meta-learning or learning-to-learn (which subsumes transfer learning as a special case). While we do not make any novel contributions with regards to this connection, we are happy to include a discussion of the connection with meta/transfer learning.
>
>
>
>
> **Interpreting experimental data**. You are correct, the dark blue in Figure 3 is lower accuracy and light yellow is higher accuracy, we will explicitly include a legend to make this clearer. Regarding the observation of multiple optima with Iris data, this corresponds to the fact that the average accuracy is a non-convex function of the splitting criterion hyperparameters $\alpha,\beta$ and can have multiple maxima.
>
>
>
>
> Finally, we would like to thank the reviewer for pointing out the couple typos, which we will address in the final version.
>
>
>
>
> *References*
>
>
> [1] Balcan, M.-F., Khodak, M., Sharma, D., & Talwalkar, A. (2021). Learning-to-learn non-convex piecewise-Lipschitz functions. NeurIPS 2021.
>
> [2] Guan, Jiechao, and Hui Xiong. "Improved Regret Bounds for Non-Convex Online-Within-Online Meta Learning." ICLR 2024.

---

### Meta-Review · Area_Chair_SpRN · 2024-04-09

This paper received five reviews with three "accept", one "Weak Accept" and one "Strong Accept". The averaged overall score achieves 7.0. I Initiated and monitored discussions. All the reviewers consistently agree on the novelty and significance of the paper. All reviewers recommend accepting the paper.